# RLHF in an SFT Way: From Optimal Solution to Reward-Weighted Alignment

**Yuhao Du**[*]          *yuhaodu@link.cuhk.edu.cn*
*Shenzhen Research Institute of Big Data*
*The Chinese University of Hong Kong, Shenzhen*

**Zhuo Li**[*]          *zhuoli3@link.cuhk.edu.cn*
*Shenzhen Research Institute of Big Data*
*The Chinese University of Hong Kong, Shenzhen*

**Pengyu Cheng**[*]          *pengyucheng95@gmail.com*
*Qwen LLM Application Team, Alibaba*

**Zhihong Chen**          *zhihongc@stanford.edu*
*Stanford University*

**Yuejiao Xie**          *yuejiaoxie@link.cuhk.edu.cn*
*The Chinese University of Hong Kong, Shenzhen*

**Xiang Wan**          *wanxiang@sribd.cn*
*Shenzhen Research Institute of Big Data*

**Anningzhe Gao**[†]          *anningzhegao@gmail.com*
*Shenzhen Research Institute of Big Data*

**Reviewed on OpenReview:** *https://openreview.net/forum?id=jewBOUhFuj*

## Abstract

Reinforcement Learning from Human Feedback (RLHF) is crucial for aligning Large Language Models (LLMs) with human values. However, RLHF has been continuously challenged by its high complexity in implementation and computation consumption, specifically for *online* sampling-based methods like Proximal Policy Optimization (PPO) and Group Relative Policy Optimization (GRPO). Even with recent simplifications, such as Direct Preference Optimization (DPO) that designs an *offline* implicit reward learning objective relying on pre-collected preference datasets, the problems of over-fitting and training instability remain hindering the alignment process from the expected optimal performance. To address the existing challenges, we propose a novel simplification of RLHF from the perspective of variational inference, called **V**ariational **A**lignment with **R**e-weighting (**VAR**). Specifically, by directly minimizing the distribution gap between the learning LLM policy and the optimal solution of RLHF, we transform the alignment objective into an *offline* reward-driven re-weighted supervised fine-tuning (SFT) form, which only requires minor adjustment on the SFT loss to obtain noticeable improvement on training stability and effectiveness. In comprehensive evaluation benchmarks, our objective empowers LLMs to outperform offline alignments, demonstrating superior performance in both helpfulness and harmlessness metrics (avg. ↑ 7.16% than DPO). Meanwhile, when compared to online sampling methods, our method is also comparable even better while significantly reducing computational overhead and accelerating convergence speed (over 5× faster than GRPO), suggesting our approach

---

[*]Equal Contribution.
[†]Corresponding Author. Code is available at `https://github.com/DuYooho/VAR`.

as an efficient and effective solution in bridging the gap between efficiency and performance in LLM alignment.

# 1 Introduction

Large Language Models (LLMs) (OpenAI, 2024; Touvron et al., 2023a; Yang et al., 2024) have achieved remarkable success in extensive applications of artificial intelligence (AI), including dialogue generation (Abdullin et al., 2024), coding (Cobbe et al., 2021; Shao et al., 2024), logical reasoning (Suzgun et al., 2022), AI agents(Wu et al., 2023; Hu et al., 2025a;b), and data curation (Du et al., 2025; Li et al., 2024a; 2025a). Among the diverse LLM training techniques, Reinforcement Learning from Human Feedback (RLHF) plays a core role in ensuring the LLM generation is helpful and harmless (Ouyang et al., 2022; Rafailov et al., 2024).

In particular, RLHF first learns a reward model (RM) (Li et al., 2025b) from annotated human preferences, then trains LLMs within a reinforcement learning (RL) scheme via Proximal Policy Optimization (PPO) (Schulman et al., 2017) to optimize the expected rewards from the learned RM (Ouyang et al., 2022). Although recognized as the mainstream solution to LLM alignment (Ouyang et al., 2022; Shao et al., 2024; Touvron et al., 2023b; OpenAI, 2024; Yang et al., 2024), RLHF remains being challenged because of its expensive computational resource consumption (Cheng et al., 2023; Yuan et al., 2023) and complicated implementation (Ouyang et al., 2022; Shao et al., 2024; Yuan et al., 2023) in which multiple models (*e.g.* the learning policy, the reference, the critic model, and the reward model) are required to cooperate in the online RL training scheme. Moreover, incorporating such a complicated pipeline significantly induces training complexity and instability, leading to the difficulty of training convergence and the high risk of collapse (Song et al., 2023; Go et al., 2023). Although methods like GRPO (Shao et al., 2024) forego the critic model, instead estimating the baseline from group scores and significantly reducing training resources, its online sampling strategy still challenges the practical implementation and training speed.

Towards more stable training than online alignment, several ranking-based offline alternatives are proposed, primarily from the perspective of enlarging the likelihood margin between preferred and rejected response pairs in a contrastive approach. Direct Preference Optimization (DPO) (Rafailov et al., 2024) implicitly maximizes the difference in sampling probabilities between good and bad answers. Ethayarajh et al. (Ethayarajh et al., 2024) introduces Kahneman-Tversky Optimization (KTO) to directly maximize the utility of generations instead of maximizing the log-likelihood of preferences. While effective, these methods usually rely on the collection of preferred / rejected response pairs with high quality, which introduces a substitution data collection consumption. Instead, Advantage Leftover Launch (A-LoL) (Baheti et al., 2024) formulates the RL process at the sequence level and derives an advantage-based offline objective that exclusively utilizes preferred responses to achieve human-aligned results. However, it still relies on clipping the importance weights to ensure training stability, which prevents the optimization from reaching the true RLHF optima. Furthermore, approaches like DPO and ALoL could employ negative weights for potential dis-preferred responses, leading to an unstable training process due to the unbounded nature of loss landscape when negative weights are applied (Pal et al., 2024; Yan et al., 2024).

In this paper, we address these limitations by proposing a reward-driven variational offline alignment framework that eliminates the need for online sampling and clipping, and avoids the instability introduced by negative weights. Our approach reformulates RLHF as a variational inference problem over positive measures, ensuring a stable and well-defined optimization landscape. Specifically, starting from the closed-form optimal solution of RLHF, we directly minimize the Kullback–Leibler (KL) divergence (Kullback & Leibler, 1951) between the to-be-learned LLM and its optimal solution. The resulting loss function takes the form of an offline reward-driven weighted supervised fine-tuning (SFT) loss, where non-negative weights are derived through an exponential reward transformation. Furthermore, we introduce an efficient in-batch normalization technique to approximate the normalization term, enabling scalable and practical implementation. Experimental results demonstrate that our framework outperforms existing methods in both stability and alignment performance, providing a robust alternative to the challenges of RLHF.

## 2 Preliminary

**Reinforcement Learning from Human Feedback** is an essential approach to alignment LLMs with human values, especially from the perspectives of helpfulness and harmlessness (Ouyang et al., 2022). We denote the input query to LLMs as $\boldsymbol{x} \in \mathcal{X}$, and the model response $\boldsymbol{y} \in \mathcal{Y}$, where $\mathcal{X}, \mathcal{Y}$ are discrete enumerable sample spaces containing natural language sequences. RLHF first learns a reward model $r(\boldsymbol{x}, \boldsymbol{y})$ from a given collection of human preference data $\mathcal{D}_{\mathrm{p}} = \{(\boldsymbol{x}, \boldsymbol{y}_w, \boldsymbol{y}_l)\}$, where $\boldsymbol{x}$ is a user input prompt, $\boldsymbol{y}_w, \boldsymbol{y}_l$ are the preferred and rejected responses selected by annotators, respectively. To learn a representative RM, following Bradley-Terry (Bradley & Terry, 1952) objective is usually utilized:

$$-\mathbb{E}_{(\boldsymbol{x}, \boldsymbol{y}_w, \boldsymbol{y}_l) \sim \mathcal{D}_{\mathrm{p}}} \Big[ \log \sigma(r(\boldsymbol{x}, \boldsymbol{y}_w) - r(\boldsymbol{x}, \boldsymbol{y}_l)) \Big], \tag{1}$$

where $\sigma(\cdot)$ is the Sigmoid function. Intuitively, Equation 1 induces $r(\boldsymbol{x}, \boldsymbol{y})$ to assign a higher reward score to the preferred response $\boldsymbol{y}_w$ than the rejected response $\boldsymbol{y}_l$ with respect to input $\boldsymbol{x}$.

With a learned RM $r(\boldsymbol{x}, \boldsymbol{y})$, RLHF optimizes the target LLM policy $\pi_\theta(\boldsymbol{y}|\boldsymbol{x})$ by maximizing the expected reward:

$$\mathbb{E}_{\boldsymbol{x} \sim \mathcal{D}, \boldsymbol{y} \sim \pi_\theta(\boldsymbol{y}|\boldsymbol{x})}[r(\boldsymbol{x}, \boldsymbol{y})] - \beta \mathrm{KL}[\pi_\theta \| \pi_{\mathrm{ref}}], \tag{2}$$

where $\mathrm{KL}[\pi_\theta \| \pi_{\mathrm{ref}}]$ is the KL divergence (Kullback & Leibler, 1951) between the training policy $\pi_\theta(\boldsymbol{y}|\boldsymbol{x})$ with a reference model $\pi_{\mathrm{ref}}(\boldsymbol{y}|\boldsymbol{x})$ to prevent $\pi_\theta(\boldsymbol{y}|\boldsymbol{x})$ from the degeneration and preserve the generation diversity. $\beta > 0$ is a hyper-parameter to re-weight the expected reward and the KL regularization term.

To solve the RLHF objective in Equation 2, Proximal Policy Optimization (PPO) (Schulman et al., 2017) has been recognized as the mainstream optimization algorithm (Rafailov et al., 2024). However, as mentioned in Section 1, PPO suffers from training instability and high complexity in computation and implementation (Yuan et al., 2023; Cheng et al., 2023). Therefore, many of recent works have been proposed to simplify and improve the original PPO algorithm. Rafailov et al. (Rafailov et al., 2024) theoretically demonstrate that Equation 2 has a closed-form solution:

$$\pi^*(\boldsymbol{y}|\boldsymbol{x}) = \frac{1}{Z(\boldsymbol{x})} \pi_{\mathrm{ref}}(\boldsymbol{y}|\boldsymbol{x}) \exp\left(\frac{1}{\beta} r(\boldsymbol{x}, \boldsymbol{y})\right), \tag{3}$$

where $Z(\boldsymbol{x}) = \mathbb{E}_{\boldsymbol{y} \sim \pi_{\mathrm{ref}}(\boldsymbol{y}|\boldsymbol{x})}[\exp(\frac{1}{\beta} r(\boldsymbol{x}, \boldsymbol{y}))]$ is the denominator that normalizes the conditional distribution. Based on the relation between the optimal policy $\pi^*(\boldsymbol{y}|\boldsymbol{x})$ and the RM $r(\boldsymbol{x}, \boldsymbol{y})$, Rafailov et al. (Rafailov et al., 2024) convert the RM learning objective Equation 1 to an optimal policy learning loss named Direct Preference Optimization (DPO):

$$-\mathbb{E}_{\mathcal{D}_{\mathrm{p}}} \left[ \log \sigma \left( \beta \log \frac{\pi_\theta(\boldsymbol{y}_w|\boldsymbol{x})}{\pi_{\mathrm{ref}}(\boldsymbol{y}_w|\boldsymbol{x})} - \beta \log \frac{\pi_\theta(\boldsymbol{y}_l|\boldsymbol{x})}{\pi_{\mathrm{ref}}(\boldsymbol{y}_l|\boldsymbol{x})} \right) \right]. \tag{4}$$

Baheti et al. (Baheti et al., 2024) adopt the PPO objective into an offline scheme by using importance sampling and converting the expectation of $\pi_\theta(\boldsymbol{y}|\boldsymbol{x})$ to the expectation of $\pi_{\mathrm{ref}}(\boldsymbol{y}|\boldsymbol{x})$, then propose Advantage-Leftover-Lunch (A-LoL) gradient estimation:

$$-\mathbb{E}_{\boldsymbol{x} \sim \mathcal{D}, \boldsymbol{y} \sim \pi_{\mathrm{ref}}(\boldsymbol{y}|\boldsymbol{x})} \left[ \hat{A}^{\pi_{\mathrm{ref}}} \cdot \frac{\pi_\theta(\boldsymbol{y}|\boldsymbol{x})}{\pi_{\mathrm{ref}}(\boldsymbol{y}|\boldsymbol{x})} \cdot \nabla_\theta \log \pi_\theta(\boldsymbol{y}|\boldsymbol{x}) \right], \tag{5}$$

where $\hat{A}^{\pi_{\mathrm{ref}}}$ is the estimated advantage value (Schulman et al., 2016) with respect to $\pi_{\bar{\theta}}$, also calculated offline.

**Variational Methods** provide a principled framework for approximating unknown probability distributions by leveraging optimization over a family of tractable parameterized distributions (Kingma & Welling, 2022). The fundamental idea of variational methods is to reformulate probabilistic inference as a functional optimization problem. More specifically, in the domain of machine learning, the goal is to find a surrogate distribution $q_\theta(\boldsymbol{y})$ from a parameterized family $\mathcal{Q} = \{q_\theta | \theta \in \Theta\}$, so that $q_\theta(\boldsymbol{y})$ can best approximates the

target unknown distribution $p(\boldsymbol{y})$. For example, in variational inference (Blei et al., 2017), the unknown distribution $p(\boldsymbol{y})$ is approximated by minimizing the *inverse* KL divergence between $q_\theta(\boldsymbol{y})$ and $p(\boldsymbol{y})$:

$$\min_{q_\theta \in \mathcal{Q}} \mathrm{KL}[q_\theta(\boldsymbol{y}) \| p(\boldsymbol{y})] = \min_{\theta \in \Theta} \mathbb{E}_{\boldsymbol{y} \sim q_\theta(\boldsymbol{y})}[\log \frac{q_\theta(\boldsymbol{y})}{p(\boldsymbol{y})}]. \tag{6}$$

Variational inference has been widely applied in various models such as variational autoencoders (VAEs) (Doersch, 2016), Bayesian Neural Networks (BNNs) (Graves, 2011), and Latent Dirichlet Allocation (LDA) (Blei et al., 2003). Besides, the *forward* KL divergence $\mathrm{KL}[p \| q_\theta] = \mathbb{E}_{\boldsymbol{y} \sim p(\boldsymbol{y})}[\log \frac{p(\boldsymbol{y})}{q_\theta(\boldsymbol{y})}]$ has also been recognized as an effective objective for variational methods, and has been applied in multiple machine learning scenarios such as f-divergence optimization (Namkoong & Duchi, 2016), reinforcement learning (Schulman et al., 2017), and Markov Chain Monte Carlo (MCMC) (Salimans et al., 2015).

## 3 Methodology

### 3.1 Variational LLM Alignment

We consider the RLHF task from a novel perspective of variational methods. Instead of optimizing the learning policy $\pi_\theta(\boldsymbol{y}|\boldsymbol{x})$ using the objective in Equation 2, we can minimize the forward $\mathrm{KL}[\pi^* \| \pi_\theta]$ between the optimal policy $\pi^*(\boldsymbol{y}|\boldsymbol{x})$ and the learning policy $\pi_\theta(\boldsymbol{y}|\boldsymbol{x})$:

$$\mathrm{KL}[\pi^* \| \pi_\theta] = \mathbb{E}_{\pi^*}\left[\log \frac{\pi^*(\boldsymbol{y}|\boldsymbol{x})}{\pi_\theta(\boldsymbol{y}|\boldsymbol{x})}\right] = -H(\pi^*) - \mathbb{E}_{\pi^*}[\log \pi_\theta(\boldsymbol{y}|\boldsymbol{x})], \tag{7}$$

where $H(\cdot)$ is the entropy of $\pi^*$ and is a constant with respect to the model parameters $\theta$. For conciseness, we adopt the expectation form of the KL divergence and let $\mathbb{E}_\pi$ denote $\mathbb{E}_{y \sim \pi(\boldsymbol{y}|\boldsymbol{x})}$. As a result, the approximation of $\pi^*$ under minimizing KL divergence can be achieved by:

$$\min_\theta \mathrm{KL}[\pi^* \| \pi_\theta] = \max_\theta \mathbb{E}_{\pi^*}[\log \pi_\theta(\boldsymbol{y}|\boldsymbol{x})]. \tag{8}$$

Using importance sampling (Goertzel, 1950; Kahn & Harris, 1951; Kloek & Van Dijk, 1978), which effectively approximates an unknown distribution with a known one, we can rearrange Equation 8 to obtain the following objective by incorporating the closed-form solution of $\pi^*$ in Equation 3:

$$\mathbb{E}_{\pi^*}[\log \pi_\theta(\boldsymbol{y}|\boldsymbol{x})] = \mathbb{E}_{\pi_{\mathrm{ref}}}\left[\frac{\pi^*(\boldsymbol{y}|\boldsymbol{x})}{\pi_{\mathrm{ref}}(\boldsymbol{y}|\boldsymbol{x})}\log \pi_\theta(\boldsymbol{y}|\boldsymbol{x})\right] = \mathbb{E}_{\pi_{\mathrm{ref}}}\left[\underbrace{\frac{\exp(\frac{1}{\beta}r(\boldsymbol{x},\boldsymbol{y}))}{Z(\boldsymbol{x})}}_{\text{Reward-driven Weight}} \cdot \underbrace{\log \pi_\theta(\boldsymbol{y}|\boldsymbol{x})}_{\text{Log-likelihood}}\right] =: \mathcal{L}_{\mathrm{VAR}}.$$

This overall objective contains two parts: a reward-driven weight and the log-likelihood of $\pi_\theta$ on the samples from the reference $\pi_{\mathrm{ref}}$, which reform the loss as a re-weight SFT loss. We call this novel alignment objective as **V**ariational **A**lignment with **R**e-weighting (VAR). However, $Z(\boldsymbol{x}) = \mathbb{E}_{\boldsymbol{y} \sim \pi_{\mathrm{ref}}(\boldsymbol{y}|\boldsymbol{x})}[\exp(\frac{1}{\beta}r(\boldsymbol{x},\boldsymbol{y}))]$ as the partition function remains unknown. Therefore, in the following subsection, we discuss the estimation of $Z(\boldsymbol{x})$ within each training micro-batch.

### 3.2 Batch-wise Partition Estimation

Equation 9 implies that the key challenge in effectively approximating $\pi^*$ through a parameterized model $\pi_\theta$ lies in the computation of $Z(\boldsymbol{x})$. However, estimating $Z(\boldsymbol{x})$ involves summing over all possible outputs $y$ for a given $x$, which can be computationally expensive, as mentioned in previous work (Rafailov et al., 2024). To avoid directly computing $Z(\boldsymbol{x})$, some alignment methods adopt policy gradient algorithms (e.g., REINFORCE (Williams, 1992) and PPO (Schulman et al., 2017)) that optimize $\pi_\theta$ without explicitly normalizing over all outputs. Hence, with a batch of sample pairs $\mathcal{B} = \{\boldsymbol{x}_i, \boldsymbol{y}_i\}_{i=1}^B$, $\boldsymbol{y}_i \sim \pi_{\mathrm{ref}}(\cdot|\boldsymbol{x}_i)$, we approximate each

---

**Algorithm 1:** VAR training within a batch

---

**Input** : Batch $\mathcal{B} = \{(\boldsymbol{x}_i, \boldsymbol{y}_i)\}_{i=1}^B$, policy $\pi_\theta(\boldsymbol{y}|\boldsymbol{x})$, reference $\pi_{\text{ref}}(\boldsymbol{y}|\boldsymbol{x})$, reward model $r(\boldsymbol{x}, \boldsymbol{y})$.

**1 for** $i \leftarrow 1$ **to** $B$ **do**

**2**     Compute policy logit $\log \pi_\theta(\boldsymbol{y}_i|\boldsymbol{x}_i)$ for the given input–output pair.

**3**     Evaluate reference logits $\{\pi_{\text{ref}}(\boldsymbol{y}_j|\boldsymbol{x}_i)\}_{j=1}^B$ over all candidates in the batch.

**4**     Obtain rewards $\{r(\boldsymbol{x}_i, \boldsymbol{y}_j)\}_{j=1}^B$ for each candidate given $\boldsymbol{x}_i$.

**5**     Calculate $\hat{Z}(\boldsymbol{x}_i) = \frac{1}{B} \sum_{j=1}^B \exp(\frac{1}{\beta} r(\boldsymbol{x}_i, \boldsymbol{y}_j)) \cdot \pi_{\text{ref}}(\boldsymbol{y}_j|\boldsymbol{x}_i)$ as the reward-weighted normalization term.

**6**     Compute $\hat{\mathcal{L}}_i = \frac{1}{\hat{Z}(\boldsymbol{x}_i)} \exp(\frac{1}{\beta} r(\boldsymbol{x}_i, \boldsymbol{y}_i)) \cdot \log \pi_\theta(\boldsymbol{y}_i|\boldsymbol{x}_i)$ as the per-sample loss term for the $i$-th example.

**7 end**

**8** Update $\pi_\theta$ with $\hat{L}_{\text{VAR}}(\mathcal{B}) = \frac{1}{B} \sum_{i=1}^B \hat{\mathcal{L}}_i$ via gradient descent.

---

$Z(\boldsymbol{x}_i)$ using Monte Carlo estimation (Salimans et al., 2015):

$$Z(\boldsymbol{x}_i) = \mathbb{E}_{\boldsymbol{y} \sim \pi_{\text{ref}}(\boldsymbol{y}|\boldsymbol{x}_i)}[\exp(\frac{1}{\beta} r(\boldsymbol{x}_i, \boldsymbol{y}))] = \sum_{\boldsymbol{y} \in \mathcal{Y}} \exp(\frac{1}{\beta} r(\boldsymbol{x}_i, \boldsymbol{y})) \cdot \pi_{\text{ref}}(\boldsymbol{y}|\boldsymbol{x}_i)$$

$$\approx \frac{1}{B} \sum_{j=1}^B \exp(\frac{1}{\beta} r(\boldsymbol{x}_i, \boldsymbol{y}_j)) \cdot \pi_{\text{ref}}(\boldsymbol{y}_j|\boldsymbol{x}_i) =: \hat{Z}(\boldsymbol{x}_i). \tag{9}$$

When the batch size $B$ is large enough, we can collect diverse $\boldsymbol{y}_j \sim \pi_{\text{ref}}(\cdot|\boldsymbol{x}_i)$ and assume that $\{\boldsymbol{y}_j\}_{j=1}^B$ are uniformly distributed in sample space $\mathcal{Y}$. With the above estimation, we can calculate the objective in equation 9 with a batch $\hat{\mathcal{L}}_{\text{VAR}}(\mathcal{B}) =$

$$\frac{1}{B} \sum_{i=1}^B \left[ \frac{\exp(\frac{1}{\beta} r(\boldsymbol{x}_i, \boldsymbol{y}_i))}{\hat{Z}(\boldsymbol{x}_i)} \cdot \log \pi_\theta(\boldsymbol{y}_i|\boldsymbol{x}_i) \right]. \tag{10}$$

In conclusion, we summarize the algorithm process of our in-batch estimation and loss computation in Algorithm 1.

### 3.3 Comparison with Previous Methods

Existing preference alignment methods exhibit two fundamental limitations. First, clipping-based approaches like PPO (Schulman et al., 2017):

$$\mathbb{E}_{\boldsymbol{x} \sim \mathcal{D}, \boldsymbol{o} \sim \pi_{\text{old}}(\boldsymbol{o}|\boldsymbol{x})} \frac{1}{|\boldsymbol{o}|} \sum_{t=1}^{|\boldsymbol{o}|} \min \left[ \frac{\pi_\theta(\boldsymbol{o}_t|\boldsymbol{x}, o_{<t})}{\pi_{\text{old}}(\boldsymbol{o}_t|\boldsymbol{x}, o_{<t})} A_t, \text{clip} \left( \frac{\pi_\theta(\boldsymbol{o}_t|\boldsymbol{x}, o_{<t})}{\pi_{\text{old}}(\boldsymbol{o}_t|\boldsymbol{x}, o_{<t})}, 1 - \epsilon, 1 + \epsilon \right) A_t \right], \tag{11}$$

and A-LoL (Baheti et al., 2024) (ref to equation 5) bound the importance ratio $\pi_\theta(\boldsymbol{y}|\boldsymbol{x})/\pi_{\text{ref}}(\boldsymbol{y}|\boldsymbol{x})$ within the interval $[1 - \epsilon, 1 + \epsilon]$. This flattens the reward distinctions between responses with similar values. For instance, when two responses have rewards $R_1 = 100$ and $R_2 = 99$, clipped methods assign nearly identical probabilities ($\sim 1/2$), failing to resolve fine-grained preferences (detailed analysis in Appendix A.1).

Second, existing methods such as DPO and A-LoL may encounter optimization instabilities caused by imbalanced or incorrect gradient dynamics. In particular, the gradient descent on dis-preferred responses can become pathologically strong, leading to an unstable optimization landscape. Consider the general weighted SFT objective:

$$\mathcal{L}_{\text{W-SFT}} = -\mathbb{E}\left[ w(\boldsymbol{x}, \boldsymbol{y}) \log \pi_\theta(\boldsymbol{y}|\boldsymbol{x}) \right]. \tag{12}$$

When $w(\boldsymbol{x}, \boldsymbol{y})$ takes negative values for dis-preferred responses, the loss becomes unbounded below. Minimizing the loss corresponds to maximizing $w(\boldsymbol{x}, \boldsymbol{y}) \log \pi_\theta(\boldsymbol{y}|\boldsymbol{x})$. For negative weights ($w < 0$), this reduces to minimizing $\log \pi_\theta(\boldsymbol{y}|\boldsymbol{x})$, creating a non-compact optimization landscape. While perfect performance

$(\log \pi_\theta(\boldsymbol{y}|\boldsymbol{x}) \to 0)$ is theoretically achievable, it is practically unreachable (Gao et al., 2023) (detailed analysis in Appendix A.2).

Our key insight is that *reward-driven alignment should operate in the space of positive measures*. We therefore propose a variational method that naturally induces non-negative weights through exponential reward transformation:

$$w(\boldsymbol{x}, \boldsymbol{y}) \propto \pi_{\text{ref}}(\boldsymbol{y}|\boldsymbol{x}) \exp(r(\boldsymbol{x}, \boldsymbol{y})/\beta) > 0. \tag{13}$$

This construction guarantees that the loss landscape has well-defined minima bounded by the reference policy's support. By reformulating RLHF as variational inference over positive measures, we achieve stable optimization without artificial clipping or negative weighting.

## 4 Experiments

To evaluate the effectiveness of our proposed approach, we conducted experiments under two primary settings: (1) the Helpful and Harmless Assistant Task (HHA) (Bai et al., 2022; Ganguli et al., 2022); and (2) generative benchmarks, including MMLU (Hendrycks et al., 2020), HumanEval (Chen et al., 2021), BigBench-Hard (Srivastava et al., 2023), and GSM8k (Cobbe et al., 2021).

### 4.1 HHA Settings

**Dataset** Our primary experiment utilizes the HHA dataset, which consists of user-assistant conversations paired with model-generated responses labeled as "chosen" or "rejected" based on human preferences. This dataset is divided into four subsets: (1) Harmless-base, containing red-teaming conversations designed to elicit harmful responses; (2) Helpful-base, (3) Helpful-online, and (4) Helpful-rejection, which focus on advice- and assistance-seeking interactions. We evaluate our method using the test sets of these subsets, comprising a total of 8.2K test conversations annotated with human preference labels.

For model training, we employ the OffsetBias (Park et al., 2024) dataset, a preference dataset similar to HHA. We utilize OffsetBias because our method explicitly relies on reward scores during training. We observed that directly applying the HHA training set and its corresponding reward models often results in inappropriate reward scores, such as instances where the chosen response receives a lower reward score than the rejected response. This issue significantly compromises the effectiveness of our method. In contrast, OffsetBias addresses six common types of biases, including length bias—where models tend to assign higher scores to longer sequences—that can mislead reward models into assigning inaccurate reward scores. By mitigating these biases, OffsetBias provides more robust and reliable reward scores, making it better suited for training our model effectively. For all HHA experiments, we use the full training set of OffsetBias, which consists of 8.5K samples.

**Models** To evaluate the scalability of our method, we conducted experiments on two model collections: Llama-{1B, 3B, 8B, 13B}[1] (Dubey et al., 2024) and Qwen2.5-{0.5B, 1.5B, 3B, 7B, 14B, 32B} (Yang et al., 2024). Specifically, we benchmark our method against DPO across all models and consider two RL training settings: 1) starting from the pre-trained (base) model; (2) starting from the SFT model. For the reward model, we employ a popular OffsetBiasRM (Park et al., 2024) that is trained on the OffsetBias preference dataset. OffsetBiasRM is designed to provide more accurate reward scores by addressing common biases, making it more suitable for our experiments.

**Evaluation** By following previous work (Rafailov et al., 2024; Baheti et al., 2024), we adopt two popular evaluation strategies: 1) **Reward Score**: A higher reward score usually indicates more useful and helpful response with respective to the input. Specifically, we use the OffsetBiasRM reward model (Park et al., 2024) to calculate reward scores for sequences generated by the aligned models on the HHA test set. Additionally, we also evaluate reward scores on the split OffsetBias evaluation set to assess the in-distribution ability of the models. 2) **Pairwise Winrate Score**: Following the common practice of LLM-as-a-judge, we evaluate model outputs from the HHA test set using the GPT-4 judge with its standardized prompt template from

---

[1]To ensure the use of the most updated models, we selected Llama3.2-{1B, 3B}, Llama3.1-8B, and Llama2-13B.

Table 1: Reward scores obtained by aligning Llama model series using the OffsetBias training set and evaluating on the four subsets of the HHA benchmark. Additionally, we report the test reward on the split test set of OffsetBias. "Avg. Helpful" denotes the average reward across Helpful-base, Helpful-online, and Helpful-rejection, while "Avg. All" represents the average reward across all four subsets of HHA.

| Method | | Harmless ↑ base | Helpful ↑ base | online | rejection | Avg. Helpful ↑ | Avg. All ↑ | OffsetBias ↑ |
|---|---|---|---|---|---|---|---|---|
| | Base | 37.03 | 20.51 | 24.04 | 21.93 | $22.16_{\pm 1.34}$ | $25.88_{\pm 1.59}$ | 21.00 |
| | DPO | 45.50 | 44.45 | 47.07 | 45.61 | $45.71_{\pm 0.16}$ | $45.66_{\pm 0.08}$ | 37.31 |
| Llama3.2-1B | VAR | 52.48 | 57.35 | 60.58 | 59.38 | $59.10_{\pm 0.24}$ | $57.44_{\pm 0.14}$ | 56.81 |
| | SFT+DPO | 56.43 | 64.65 | 64.95 | 65.90 | $65.16_{\pm 0.43}$ | $62.98_{\pm 0.30}$ | 59.09 |
| | SFT+VAR | **60.19** | **65.96** | **68.94** | **68.27** | $\mathbf{67.72}_{\pm 0.11}$ | $\mathbf{65.84}_{\pm 0.07}$ | **61.97** |
| | Base | 35.05 | 26.50 | 31.15 | 28.60 | $28.75_{\pm 0.38}$ | $30.33_{\pm 0.14}$ | 26.61 |
| | DPO | 53.71 | 59.38 | 60.04 | 60.55 | $59.99_{\pm 0.05}$ | $58.42_{\pm 0.09}$ | 53.94 |
| Llama3.2-3B | VAR | 57.97 | 60.23 | 64.92 | 62.92 | $62.69_{\pm 0.04}$ | $61.51_{\pm 0.08}$ | 60.88 |
| | SFT+DPO | **64.00** | **69.44** | 71.01 | **71.56** | $\mathbf{70.67}_{\pm 0.08}$ | $\mathbf{69.00}_{\pm 0.05}$ | **63.81** |
| | SFT+VAR | **64.00** | 67.93 | **71.32** | 70.83 | $70.02_{\pm 0.22}$ | $68.52_{\pm 0.12}$ | 63.72 |
| | Base | 38.73 | 34.74 | 39.96 | 37.30 | $37.33_{\pm 0.65}$ | $37.68_{\pm 0.46}$ | 30.42 |
| | DPO | 56.17 | 59.89 | 61.02 | 60.86 | $60.59_{\pm 0.02}$ | $59.48_{\pm 0.03}$ | 51.38 |
| Llama3.1-8B | VAR | 57.18 | 61.13 | 65.57 | 64.33 | $63.68_{\pm 0.20}$ | $62.06_{\pm 0.11}$ | 63.91 |
| | SFT+DPO | 62.38 | **69.16** | 70.00 | 70.61 | $\mathbf{69.93}_{\pm 0.07}$ | $68.04_{\pm 0.11}$ | 60.88 |
| | SFT+VAR | **63.24** | 68.13 | **71.13** | **70.78** | $70.01_{\pm 0.47}$ | $\mathbf{68.32}_{\pm 0.51}$ | **65.75** |
| | Base | 33.06 | 27.39 | 29.53 | 28.36 | $28.43_{\pm 0.04}$ | $29.59_{\pm 0.13}$ | 27.05 |
| | DPO | 50.52 | 50.01 | 53.68 | 52.24 | $51.98_{\pm 0.15}$ | $51.61_{\pm 0.14}$ | 51.75 |
| Llama2-13B | VAR | 58.45 | 58.94 | 62.94 | 61.89 | $61.26_{\pm 0.24}$ | $60.56_{\pm 0.25}$ | 61.44 |
| | SFT+DPO | 55.19 | 59.90 | 60.61 | 61.26 | $60.59_{\pm 0.25}$ | $59.24_{\pm 0.19}$ | 59.09 |
| | SFT+VAR | **61.29** | **63.27** | **66.07** | **65.75** | $\mathbf{65.03}_{\pm 0.15}$ | $\mathbf{64.09}_{\pm 0.09}$ | **62.59** |

MT-Bench (Zheng et al., 2023)[2]. To alleviate potential positional bias, we present the responses of two models to the judge in two different orders and compare their scores. A model is considered to win only if it does not lose in both orderings. Specifically, we define: *Wins*: Outperforms in both orderings or wins in one and ties in the other. *Tie*: Ties in both orderings or wins in one and loses in the other. *Loses*: Lags in both orderings or ties in one and loses in the other.

**Implementation Details** For all comparisons, we ensure consistent settings between our method and DPO. We select learning rates of 1e-5 and 5e-6, employing the AdamW optimizer with a cosine learning rate scheduler. For the Winrate evaluation, we randomly sample 99 instances from each of the four subsets of the HHA test set and conduct experiments using three different random seeds to compute the mean and standard error. To mitigate potential positional bias in GPT-4's preferences, we randomly shuffle the positions of model-generated sequences and the SFT target during evaluation using `gpt-4o-2024-11-20`. For the Reward evaluation, we use the entire HHA test set, evaluating across three different random seeds to compute the mean and standard error. We employ generation parameters $\tau = 0.8$, top_p $= 0.9$ and top_k $= 50$ for all the generations. All models are trained on 4×A100-80GB GPUs, with Llama3-8B and Qwen2.5-14B utilizing 8×A100-80GB. Due to resource constraints, Qwen2.5-32B is trained using 4-bit quantization (bnb-4bit)[3] across 4 nodes, each with 8×A100-80GB GPUs.

## 4.2 HHA Results

**Reward Evaluation** Table 1 present the reward scores on the HHA test sets. *DPO* and *VAR* denote models trained directly from the *Base* model (pre-trained only), while *SFT+* refers to models first fine-tuned via SFT and then further fine-tuned on the SFT model. From Table 1, we observe that our method outperforms DPO in both *Avg. Helpful* and *Avg. All* across all Llama models for the *base* version, as well as for the *SFT+* version, except for Llama3.2-3B, where it shows a marginal decrease of around 0.5% compared to DPO. Additionally, our method achieves comparable results whether trained directly from the base model or the SFT model, particularly for larger LLMs such as Llama3.1-8B and Llama2-13B, whereas DPO struggles to achieve strong results when starting from the base model. Moreover, our method achieves performance comparable to the RLHF objective in a single training step, resembling the simplicity and

---

[2]Note that we only adopt the MT-Bench judging protocol but do not use the MT-Bench benchmark itself.
[3]https://huggingface.co/docs/bitsandbytes

Table 2: Reward scores obtained by aligning Qwen2.5 model series using the OffsetBias training set and evaluating on the four subsets of the HHA benchmark. Additionally, we report the test reward on the split test set of OffsetBias. "Avg. Helpful" denotes the average reward across Helpful-base, Helpful-online, and Helpful-rejection, while "Avg. All" represents the average reward across all four subsets of HHA.

| Method | | Harmless ↑ base | Helpful ↑ base | online | rejection | Avg. Helpful ↑ | Avg. All ↑ | OffsetBias ↑ |
|---|---|---|---|---|---|---|---|---|
| Qwen2.5-0.5B | Base | 33.03 | 25.44 | 30.86 | 26.94 | $27.75_{\pm 0.30}$ | $29.06_{\pm 0.14}$ | 40.38 |
| | DPO | 55.21 | 55.50 | 56.24 | 56.75 | $56.17_{\pm 0.53}$ | $55.93_{\pm 0.45}$ | 53.44 |
| | VAR | 55.22 | 58.09 | 62.32 | 60.38 | $60.26_{\pm 0.21}$ | $59.00_{\pm 0.07}$ | 59.50 |
| | SFT+DPO | 56.42 | 58.02 | 60.38 | 59.91 | $59.44_{\pm 0.03}$ | $58.68_{\pm 0.02}$ | 55.88 |
| | SFT+VAR | **58.22** | **61.72** | **63.58** | **63.56** | $\mathbf{62.95}_{\pm 0.14}$ | $\mathbf{61.77}_{\pm 0.12}$ | **60.63** |
| Qwen2.5-1.5B | Base | 35.01 | 26.18 | 32.11 | 28.13 | $28.81_{\pm 0.12}$ | $30.36_{\pm 0.32}$ | 26.52 |
| | DPO | 53.40 | 57.00 | 57.38 | 57.85 | $57.41_{\pm 0.17}$ | $56.41_{\pm 0.13}$ | 56.63 |
| | VAR | 61.33 | 64.72 | 68.87 | 68.01 | $67.20_{\pm 0.09}$ | $65.73_{\pm 0.05}$ | 64.75 |
| | SFT+DPO | 54.51 | 61.13 | 61.54 | 62.26 | $61.64_{\pm 0.32}$ | $59.86_{\pm 0.28}$ | 57.63 |
| | SFT+VAR | **62.76** | **66.07** | **69.13** | **68.78** | $\mathbf{67.99}_{\pm 0.13}$ | $\mathbf{66.69}_{\pm 0.03}$ | **65.88** |
| Qwen2.5-3B | Base | 47.07 | 34.27 | 41.86 | 36.61 | $37.58_{\pm 0.14}$ | $39.96_{\pm 0.18}$ | 45.06 |
| | DPO | 60.58 | 63.37 | 63.84 | 64.90 | $64.03_{\pm 0.53}$ | $63.17_{\pm 0.51}$ | 58.97 |
| | VAR | 63.30 | 66.29 | 70.07 | 69.32 | $68.56_{\pm 0.08}$ | $67.24_{\pm 0.05}$ | **66.06** |
| | SFT+DPO | 54.61 | 58.45 | 58.69 | 60.19 | $59.11_{\pm 0.17}$ | $57.98_{\pm 0.17}$ | 52.63 |
| | SFT+VAR | **65.15** | **67.86** | **71.24** | **70.78** | $\mathbf{69.96}_{\pm 0.06}$ | $\mathbf{68.75}_{\pm 0.05}$ | 65.63 |
| Qwen2.5-7B | Base | 42.49 | 42.02 | 47.96 | 44.21 | $44.73_{\pm 0.53}$ | $44.17_{\pm 0.28}$ | 54.09 |
| | DPO | 61.51 | 66.09 | 66.50 | 67.28 | $66.62_{\pm 0.18}$ | $65.35_{\pm 0.10}$ | **65.81** |
| | VAR | 64.86 | 66.41 | **70.87** | 69.78 | $\mathbf{69.02}_{\pm 0.09}$ | $\mathbf{67.98}_{\pm 0.03}$ | 65.38 |
| | SFT+DPO | 60.78 | 62.02 | 61.62 | 62.67 | $62.10_{\pm 0.25}$ | $61.77_{\pm 0.30}$ | 56.91 |
| | SFT+VAR | **64.96** | **66.46** | 69.81 | 69.39 | $68.55_{\pm 0.11}$ | $67.65_{\pm 0.05}$ | 65.38 |
| Qwen2.5-14B | Base | 39.29 | 32.73 | 39.03 | 35.16 | $35.64_{\pm 0.08}$ | $36.55_{\pm 0.27}$ | 45.16 |
| | DPO | 55.83 | 55.63 | 59.04 | 57.61 | $57.43_{\pm 0.32}$ | $57.03_{\pm 0.26}$ | 62.94 |
| | VAR | 64.24 | 65.23 | 69.79 | 68.50 | $67.84_{\pm 0.14}$ | $66.94_{\pm 0.03}$ | 65.63 |
| | SFT+DPO | **66.97** | **67.83** | 68.16 | 68.94 | $68.31_{\pm 0.42}$ | $67.97_{\pm 0.38}$ | **67.94** |
| | SFT+VAR | 66.37 | 67.74 | **71.70** | **71.08** | $\mathbf{70.17}_{\pm 0.21}$ | $\mathbf{69.22}_{\pm 0.10}$ | 66.44 |
| Qwen2.5-32B-Int4 | Base | 38.80 | 34.36 | 39.13 | 36.78 | 36.77 | 37.27 | 38.97 |
| | DPO | 37.09 | 31.38 | 34.58 | 33.04 | 33.00 | 34.02 | 32.38 |
| | VAR | 50.03 | 45.36 | 51.77 | 47.77 | 48.30 | 48.73 | 57.69 |
| | SFT+DPO | 37.95 | 27.70 | 28.63 | 28.16 | 28.16 | 30.61 | 30.23 |
| | SFT+VAR | **53.18** | **49.07** | **55.66** | **51.90** | **52.21** | **52.45** | **59.09** |

Table 3: Winrate results for the Llama3 series instruct versions (1B, 3B, and 8B) on the HHA benchmark.

| Method | | Harmless-base ↑ A win | B win | Tie | Helpful-base ↑ A win | B win | Tie | Helpful-online ↑ A win | B win | Tie | Helpful-rejection ↑ A win | B win | Tie | Avg. Winrate ↑ All | Helpful |
|---|---|---|---|---|---|---|---|---|---|---|---|---|---|---|---|
| Llama3.2-1B | DPO | 20.2 | 59.6 | 20.2 | 49.49 | 32.32 | 18.18 | 12.46 | 71.38 | 16.16 | 34.68 | 47.47 | 17.84 | $29.21_{\pm 0.17}$ | $32.21_{\pm 0.41}$ |
| | VAR | **55.56** | 34.34 | 10.10 | **72.73** | 16.50 | 10.77 | **48.48** | 42.42 | 9.09 | **69.36** | 19.86 | 10.77 | $\mathbf{61.53}_{\pm 0.22}$ | $\mathbf{63.53}_{\pm 0.30}$ |
| Llama3.2-3B | DPO | **76.10** | 20.54 | 3.37 | 76.77 | 14.81 | 8.42 | 49.83 | 44.10 | 6.06 | 62.97 | 25.59 | 11.45 | $66.42_{\pm 0.58}$ | $63.19_{\pm 0.74}$ |
| | VAR | 63.30 | 26.26 | 10.44 | **89.90** | 3.70 | 6.40 | **61.96** | 32.66 | 5.39 | **79.13** | 12.12 | 8.75 | $\mathbf{73.57}_{\pm 0.95}$ | $\mathbf{76.99}_{\pm 0.98}$ |
| Llama3.1-8B | DPO | 42.42 | 42.42 | 15.15 | 68.69 | 19.19 | 12.12 | 33.33 | 54.21 | 12.46 | 53.20 | 31.65 | 15.15 | $49.41_{\pm 0.17}$ | $51.74_{\pm 0.11}$ |
| | VAR | **51.52** | 33.67 | 14.81 | **88.55** | 8.08 | 3.37 | **58.59** | 34.34 | 7.07 | **85.86** | 10.44 | 3.70 | $\mathbf{71.13}_{\pm 0.34}$ | $\mathbf{77.67}_{\pm 0.30}$ |

efficiency of SFT. Table 2 further demonstrates the scalability of our method across different model sizes. Our approach consistently outperforms DPO across all average reward scores on both HHA and OffsetBias, even when starting from the base model. For Qwen2.5-32B, due to limited resources, we employ 4-bit quantization for training. Nevertheless, our method maintains its advantage over DPO.

**Winrate Evaluation** Figure 1 presents the win rates evaluated by GPT-4o for answers generated by aligned models compared to the SFT targets (chosen answers in the test set) for the Llama and Qwen series. The Llama series show results consistent with the reward scores, where our method outperforms DPO across all models except Llama2-13B, which achieves comparable results. For the Qwen series, our method outperforms DPO across all Qwen models except Qwen2.5-14B, where it shows a slight decrease. The Qwen series exhibit slightly different trends compared to the reward scores, with our method starting from the base version outperforming the SFT+ version across scales from 0.5B to 14B and achieving comparable results at 32B. These findings further demonstrate that our method can achieve the RLHF objective in a single SFT-like step without the need for resource-intensive reinforcement learning.

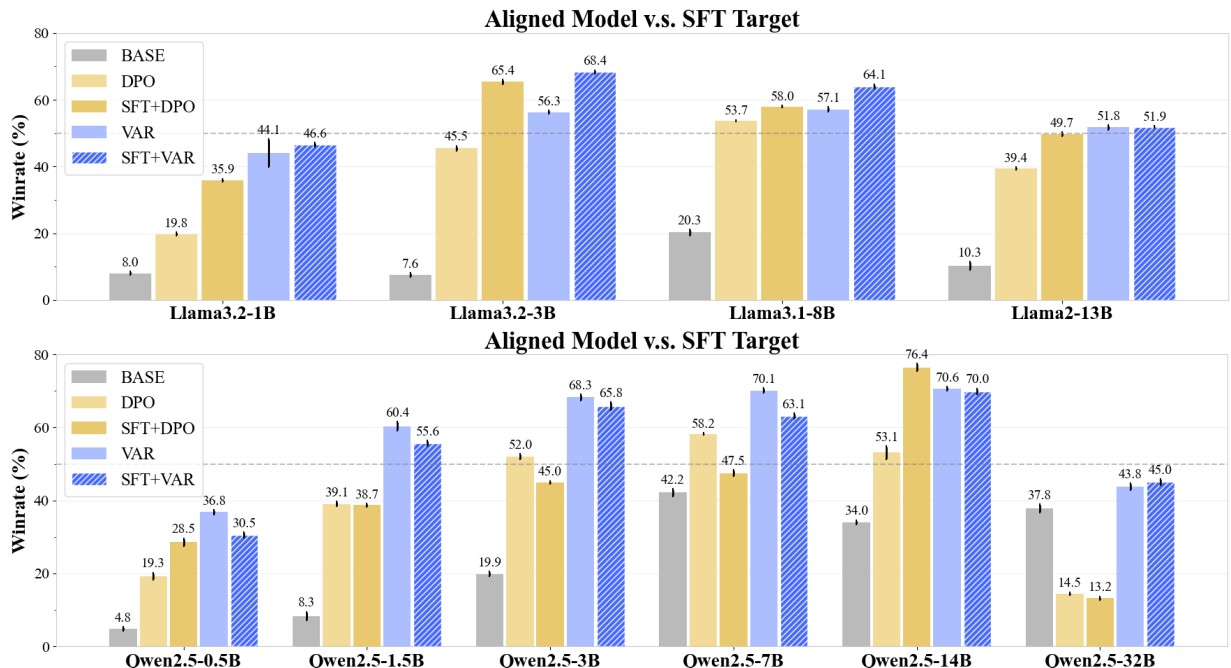

Figure 1: GPT-4 evaluation results on the HHA test set for the Llama (top) and Qwen (bottom) series, reporting average win rates, where error bars are calculated across three different random seeds.

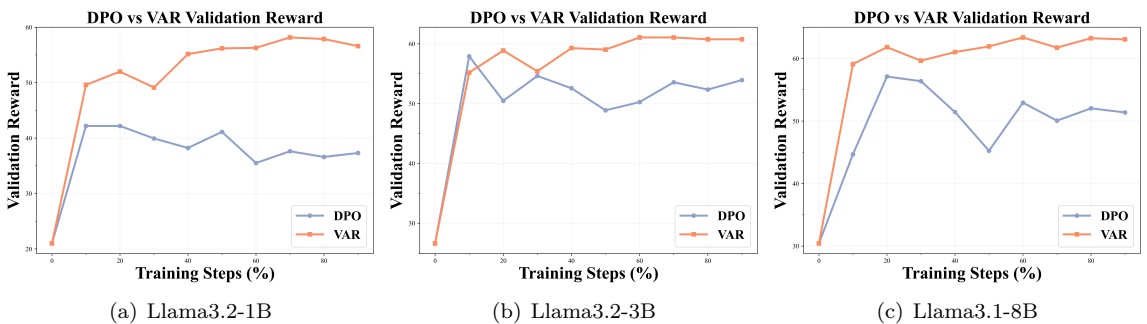

Figure 2: Average validation reward during the training process for (a) Llama3.2-1B, (b) Llama3.2-3B, and (c) Llama3.1-8B on the OffsetBias dataset, comparing DPO and our method.

Table 3 shows results on the Llama series instruct versions (i.e., models after RLHF). Our method outperforms DPO by a large margin across three scales (1B, 3B, and 8B) and consistently achieves higher win rates across all subsets of the HHA benchmark. This demonstrates the robustness of our method across models at different training stages, including those that have already undergone RLHF alignment. The results highlight the effectiveness of our approach in further refining and aligning models with human preferences, even when starting from pre-aligned instruct versions.

**Training Stability** Figure 2 illustrates the average validation reward during the training process for three Llama model collections. Comparing our method with DPO, we observe that DPO exhibits greater volatility and tends to decline from the early training steps. In contrast, our method demonstrates a gradual increase in validation reward, ultimately reaching a consistent level. This indicates that our approach is more robust over longer training steps and maintains a more stable training process compared to DPO.

**Comparison with Online Methods** We evaluate our offline VAR method against the online GRPO approach (Shao et al., 2024) using the OffsetBias training set, with results measured on the HHA test sets (Table 4). Despite being an offline method, VAR achieves

Table 4: Comparison between offline VAR and online GRPO methods on Qwen2.5-3B and Llama3.2-3B, including training time per epoch.

| Method | Harmless ↑ base | Helpful ↑ | | | Avg. Helpful ↑ | Avg. All ↑ | Train. Time ↓ |
|--------|------|------|--------|-----------|----------------|------------|---------------|
| | | base | online | rejection | | | |
| Qwen2.5-3B | 47.07 | 34.27 | 41.86 | 36.61 | 37.58 | 39.95 | – |
| +GRPO | 61.75 | **68.78** | **70.73** | **71.64** | **70.38** | **68.23** | 2h54min |
| +VAR | **63.30** | 66.29 | 70.07 | 69.32 | 68.56 | 67.25 | **42min** |
| Llama3.2-3B | 33.03 | 25.44 | 30.86 | 26.94 | 27.75 | 29.06 | – |
| +SFT+GRPO | 56.34 | 59.98 | 62.32 | 62.72 | 61.67 | 60.34 | 3h8min |
| +VAR | **57.97** | **60.23** | **64.92** | 62.92 | **62.69** | **61.51** | **37min** |

comparable performance to GRPO on Qwen2.5-3B and even outperforms it on Llama3.2-3B (where GRPO requires SFT initialization for stable training). Notably, VAR trains approximately 5× faster per epoch than GRPO, demonstrating its efficiency while maintaining competitive performance with significantly fewer computational resources.

## 4.3 Generative Benchmark

**Settings** Following prior settings (Tunstall et al., 2023; Ethayarajh et al., 2024), we utilize UltraFeedback (Cui et al., 2023) as the training dataset. UltraFeedback is a large-scale preference dataset collected from diverse sources, where multiple LLMs generate four distinct responses for each prompt. The dataset comprises 64k prompts, resulting in 256k samples. Additionally, it includes GPT-4-evaluated scores for instruction-following, truthfulness, honesty, and helpfulness. For our experiments, we sampled 10k prompts, selecting the highest average-scored samples for training SFT, OURS, and ALoL, and using the highest-worst score pairs for training DPO. For training, we utilize Llama2-7B and Qwen2.5-7B as the base models. For comparison, we benchmark our method against ALoL and DPO. As for the reward model, we employ OffsetBiasRM for both ALoL and our method, with the same setting in HHA experiments.

**Implementation Details** For all experiments, we use $4 \times 8$ A100-80GB GPUs, training with a learning rate of 5e-6 and the AdamW optimizer combined with a cosine learning rate scheduler for exactly two epochs. We evaluate the aligned models using the OpenCompass (Contributors, 2023) toolkit, with the following benchmarks: GSM8K (4-shot), MMLU (0-shot), HumanEval (0-shot), and BBH (3-shot chain-of-thought), following the default settings from OpenCompass. For GSM8K, MMLU, and BBH, we use exact match (EM) as the evaluation metric, while for HumanEval, we use pass@1.

**Generative Benchmark Results** Table 5 presents the results of different methods on Llama2-7B and Qwen2.5-7B models across four benchmarks: MMLU, GSM8k, HumanEval, and BBH. For Llama2-7B, our method consistently outperforms both DPO and ALoL across most benchmarks, achieving the highest average results. On Qwen2.5-7B, our method also demonstrates strong results, achieving the best performance on multiple benchmarks while maintaining competitive results on others, highlighting the robustness and effectiveness across various settings.

To further assess general conversational ability, we also evaluate the models on AlpacaEval 2.0 and Arena-Hard 0.1. The results are presented in Table 6. On the AlpacaEval 2.0 benchmark, DPO shows a slight advantage, achieving a 4.78% win rate (WR) and an 8.73% length-controlled (LC) win rate, compared to 4.29% and 8.26% for VAR, respectively. However, on the more challenging Arena-Hard 0.1 benchmark, our VAR method clearly outperforms DPO, scoring 10.8% to DPO's 8.6%. This suggests that the two methods may have different strengths across various conversational benchmarks.

Table 6: Conversational benchmark results on AlpacaEval 2.0 and Arena-Hard 0.1.

| Method | AlpacaEval 2.0 ↑ | | Arena-Hard 0.1 ↑ |
|--------|--------|--------|------------|
| | LC (%) | WR (%) | WR (%) |
| Base | 4.01 | 3.73 | 8.4 |
| DPO | **8.73** | **4.78** | 8.6 |
| VAR | 8.26 | 4.29 | **10.8** |

## 4.4 Output Sequence Length Analysis

We calculate the average output sequence lengths for models aligned on Llama3.1-8B across the four subsets of HHA, as shown in Figure 3. Starting from the base models, DPO generates longer sequences than our

Table 5: Benchmark comparison of different alignment methods on Llama2-7B and Qwen2.5-7B models. Results are shown for MMLU, GSM8K, HumanEval, and BBH, along with the average score across all benchmarks. Best results for each model are bolded.

| Method | | MMLU EM ↑ | GSM8k EM ↑ | HumanEval pass@1 ↑ | BBH EM ↑ | Avg. ↑ |
|---|---|---|---|---|---|---|
| Llama2-7B | Base | 37.46 | 1.90 | 3.05 | 12.77 | 13.79 |
| | DPO | 32.45 | 4.55 | 7.32 | **39.10** | 20.86 |
| | ALoL | 35.78 | 4.09 | 12.80 | 38.16 | 22.71 |
| | VAR | **38.57** | **6.67** | **14.02** | 37.56 | **24.20** |
| Qwen2.5-7B | Base | 67.13 | **86.13** | 64.63 | 29.30 | 61.80 |
| | DPO | 68.64 | 74.53 | 33.54 | 53.06 | 57.44 |
| | ALoL | 68.62 | 64.97 | 54.88 | 61.14 | 62.40 |
| | VAR | **69.11** | 74.30 | **68.90** | **61.35** | **68.42** |

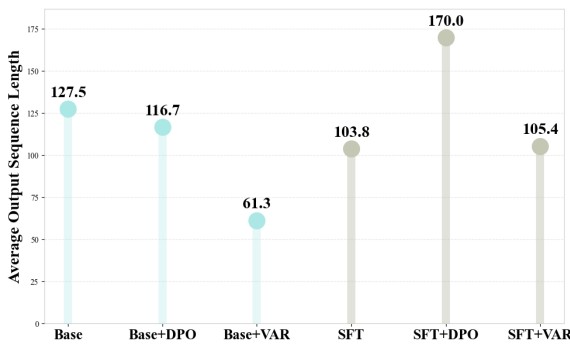

Figure 3: Distribution of output sequence lengths for aligned Llama3.1-8B on the HHA test set.

method. When starting from SFT, our models maintain output sequence lengths similar to the SFT version, while DPO produces sequences approximately twice as long as ours and the SFT version. Longer sequences tend to achieve higher reward scores and GPT-based scores (Baheti et al., 2024; Ethayarajh et al., 2024). Despite this, our method outperforms DPO in most reward evaluations and winrate evaluations.

### 4.5 Ablation Study

As in Equation 9, we estimate $Z(\boldsymbol{x}_i)$ with a micro batch, making batch size $B$ crucial for training. We conduct an ablation study using batch sizes 2, 4, and 8 under the settings in Section 4.3, with UltraFeedback as the training set. Table 7 shows the impact of different $B$ values on model performance. The model performs best at $B = 8$, but the improvement over $B = 4$ is just 0.14%, suggesting that a

Table 7: Ablation study of different batch sizes $B$ on Qwen2.5-7B model and $\pi_{\mathrm{ref}}$ sampled alternative ($8^*$).

| $B$ | MMLU EM ↑ | GSM8k EM ↑ | HumanEval pass@1 ↑ | BBH EM ↑ | Avg. ↑ |
|---|---|---|---|---|---|
| 2 | **69.59** | 72.55 | 65.24 | 62.57 | 67.49 |
| 4 | 69.09 | 75.44 | 68.29 | 60.32 | 68.28 |
| 8 | 69.11 | 74.30 | **68.90** | 61.35 | **68.42** |
| $8^*$ | 67.83 | **75.97** | 65.85 | **63.66** | 68.32 |

larger $B$ can slightly boost performance. Thus, we use $B = 8$ for all experiments. However, our method is also robust to batch-size changes, which gives satisfactory results even at $B = 2$, showing stability and suitability for resource-constrained situations.[4]

Furthermore, we compare our in-batch (offline) estimator to an alternative strategy, $\pi_{\mathrm{ref}}$ sampled ($B = 8$), which estimates $\hat{Z}(x)$ using dedicated candidates pre-sampled from $\pi_{\mathrm{ref}}$ for each prompt. As shown in the last row of Table 7, this alternative approach is not only significantly more costly (requiring a separate, large-scale pre-sampling process) but also results in slightly lower performance (68.32 Avg. vs. 68.42 Avg.). This result validates our choice of the more efficient and effective in-batch estimation strategy.

### 4.6 Analyses of $Z(x)$ Estimation

The stability of our VAR method depends heavily on getting a good estimate for the partition function $Z(x)$. Our approach uses an efficient Monte Carlo estimation based on the samples already available in the training micro-batch (as described in Equation 9 and Algorithm 1). A fair question is how reliable this estimate is, and specifically, how the micro-batch size $B$ affects its variance.

To investigate this, we ran an analysis, shown in Figure 4. We compare the variance of $\hat{Z}(x)$ under two different estimation strategies: (1) **Offline (In-Batch) Estimation (Figure 4 (a))**: This is the method we use in this work; it leverages the $B$ samples already in the micro-batch to calculate $\hat{Z}(x_i)$ for each $x_i$ in that batch. The plot clearly shows that the variance drops sharply as $B$ increases from 2 to 8. This confirms that even a modest batch size (like $B = 8$, which we used in our experiments ) provides a significantly more stable estimate than a very small one. (2) **Online Sample Estimation (Figure 4 (b))**: For comparison,

---

[4]We provide case study in Appendix C, from which we can observe our method effectively produce more comprehensive and helpful response with higher quality compared to the baseline methods.

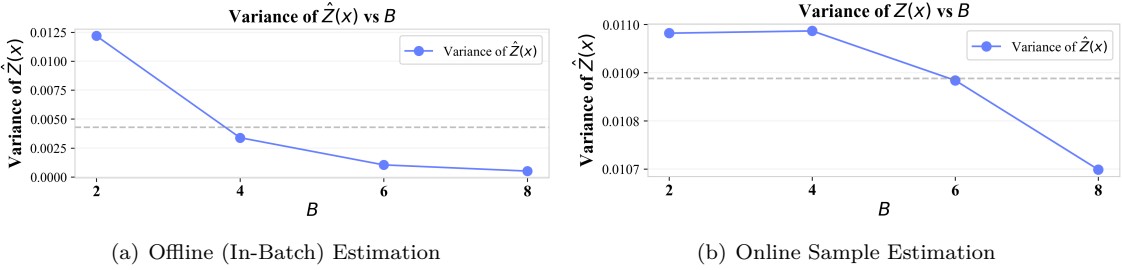

Figure 4: The variance of the partition function estimate $\hat{Z}(x)$ versus the micro-batch size $B$.

we simulated an "online" approach. Here, for each $x_i$, we would need to actively sample $B$ new responses from the reference policy $\pi_{\mathrm{ref}}$ to compute $\hat{Z}(x_i)$.

This analysis confirms our design choice. The online method is not only computationally expensive (requiring $B$ extra sampling steps per instance) but also shows a higher and less consistent variance at small batch sizes. Our in-batch (offline) method is significantly more efficient and yields a stable, low-variance estimate as the batch size increases.

### 4.7 Computational Overhead Analysis

Our method is structured to avoid adding significant overhead to the main training loop. We achieve this by integrating the costliest parts of the $\hat{Z}(x)$ estimation into the **data preprocessing step**: **(1) Preparation:** Before the main training loop starts, we pre-shuffle the training data and split it into the exact **micro-batches** (e.g., $B = 8$) that will be used for training. **(2) Reward Calculation:** We then calculate the reward $r(x_i, y_j)$ for each in-batch pair and store these values on disk.

Table 8: Computational profile comparison for a single training epoch (Llama3.2-3B, $B = 8$, 4×A100-80GB).

| Method | Wall Clock Time/Epoch ↓ | Throughput Tokens/s ↑ | Peak GPU Mem./GPU ↓ |
|---|---|---|---|
| SFT | ∼28min | ∼2945 | ∼31 GB |
| DPO | ∼32min | ∼2577 | ∼39 GB |
| **VAR** | ∼37min | ∼2228 | ∼38 GB |

Therefore, during training, we only need to perform one major extra step compared to standard SFT: a forward pass through the **reference model ($\pi_{\mathrm{ref}}$)** to obtain the logits necessary to compute $\hat{Z}(x)$ (using the pre-calculated rewards). However, the most time-consuming part of training LLMs is the **backward pass (gradient computation)**. Since our training loop does not involve running the backward pass on the reference model ($\pi_{\mathrm{ref}}$ is frozen), the minimal extra cost incurred by the single forward pass is negligible when amortized over the entire training process.

We provide a direct comparison of SFT, DPO, and VAR in Table 8, run under identical settings (Llama3.2-3B, 4×A100-80GB GPUs, $B = 8$). The results confirm our approach is highly efficient. As shown in the table, the wall clock time per epoch for VAR (37min) is close to SFT (28min) and DPO (32min). This minimal time difference is reflected in a slight reduction in throughput. For peak memory, VAR is comparable to DPO, as both must load the $\pi_{\mathrm{ref}}$.

## 5 Related Work

Aligning LLMs with human preferences has evolved from studies on RLHF, aiming to achieve human-aligned outcomes (Stiennon et al., 2020; Ouyang et al., 2022; Bai et al., 2022; Lee et al., 2023). The RLHF process typically begins with SFT, followed by further fine-tuning to maximize expected reward scores. This requires the construction of a reward model based on the Maximum Likelihood Estimation (MLE) of the BT model to provide such reward scores. This fine-tuning process is referred to as RLHF, with the PPO algorithm being the most widely applied (Schulman et al., 2017). A series of works focus on self-training, where the workflow involves sampling online data from the model and training it using a two-player min-max game between two policies (Rosset et al., 2024; Swamy et al., 2024; Chen et al., 2024). However, the use of online data in

the learning process presents significant challenges, as it requires substantial computational resources and limits training efficiency. To address these challenges, researchers have shifted their focus to offline preference alignment learning algorithms. These methods operate in a single stage and directly optimize a designed loss function to achieve optimal preference alignment based on pairwise datasets (Zhao et al., 2023; Rafailov et al., 2024; Azar et al., 2024; Ethayarajh et al., 2024; Xu et al., 2024). However, standard offline methods like DPO often rely on a fixed reference model, which can lead to optimization instability. To mitigate this, recent alternatives have proposed eliminating the reference-policy mismatch via reference-free objectives (Meng et al., 2024) or minimizing the divergence by learning a better reference model (Gorbatovski et al., 2024). Separately, to alleviate the resource-intensive nature of training, Remax (Li et al., 2024b) introduces a variance reduction method for LLMs. Most closely related to our work is ALoL (Baheti et al., 2024), which formulates the reinforcement learning process at the sequence level and derives its advantage-based offline objective. This sequence-level perspective is further supported by recent theoretical analysis linking SFT generalization to reward rectification (Wu et al., 2025), reinforcing the motivation for formulating RLHF within a supervised framework. However, unlike ALoL which relies on heuristic clipping, or methods like SimPO that discard the reference model, we formulate our approach by directly approximating the optimal solution of RLHF via variational inference, thereby achieving a more precise and stable solution.

## 6 Conclusion

This paper proposed a reward-driven variational alignment framework to address the limitations of existing RLHF methods, such as instability from negative weights and suboptimal performance due to clipping. By reformulating RLHF as a variational problem over positive measures, our approach ensures a stable optimization landscape and derives a reward-driven weighted SFT loss through KL divergence minimization. The introduction of an efficient in-batch normalization technique further enables scalable and practical implementation. Experimental results demonstrate improved alignment performance and training stability, offering a effective solution for preference alignment in LLM.

## Impact Statement

The Variational Alignment with Re-weighting (VAR) method has a significant impact on LLM alignment, providing an efficient and stable alternative to complex RLHF pipelines. While our focus is on improving alignment performance and training stability, we recognize the broader societal implications, including the risks of bias amplification, misuse of generative capabilities, and ethical concerns surrounding the automation of human-like decision-making. We emphasize the need for ongoing research to address these challenges and ensure responsible deployment of RLHF advancements.

## Acknowledgment

This work was supported by the Guangxi Key R&D Project (No. AB24010167), Guangxi Science and Technology Program under Grant No. FN2504240022, the Project (No. 20232ABC03A25), Shenzhen Longgang District Science and Technology Innovation Special Fund (No. LGKCYLWS2023018), Futian Healthcare Research Project (No.FTWS002), and Central Funds Guiding the Local Science and Technology Development Project (No. 2025ZYDF106).

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

# A   Theoretical Analysis

## A.1   Analysis of Clip Operator and Policy Distinction

The clip operator in PPO is:

$$\text{clip}(r(\theta, ref), 1 - \epsilon, 1 + \epsilon),$$

commonly used in methods like ALoL to stailize training, bounds the importance ratio $r(\theta, \text{ref}) = \frac{\pi_\theta(y_i|x)}{\pi_{\text{ref}}(y_i|x)}$ within $[1 - \epsilon, 1 + \epsilon]$. While effective in controlling gradient variance, clipping introduces bias by flattening reward distinctions between responses with similar values.

For instance, suppose for a given instruction $x$, we have a set of answers $\{y_1, y_2, ..., y_n\}$, and the loss function of R-LoL (R-LoL used here as a simple example; the only difference between A-LoL and R-LoL is replacing $r(\boldsymbol{x}, \boldsymbol{y})$ in R-LoL to $A(x, y) = r(\boldsymbol{x}, \boldsymbol{y}) - V(x)$) with clipping is:

$$\mathcal{L}_{\text{R-LoL}} = \sum_{i=1}^{n} (R(x, y_i) \cdot \text{clip}(r(\theta, ref), 1 - \epsilon, 1 + \epsilon) \log \pi_\theta(\boldsymbol{y}|\boldsymbol{x}) - \beta \log \pi_\theta(y_i|x)) \tag{14}$$

$$= \sum_{i=1}^{n} (R(x, y_i) \cdot \text{clip}(r(\theta, ref), 1 - \epsilon, 1 + \epsilon) - \beta) \log \pi_\theta(y_i|x). \tag{15}$$

When $\epsilon$ is small, we assume that the parameter $\theta$ in the function $r(\theta, ref)$ is frozen when we do the update of the policy model $\pi_\theta$, i.e., the loss function becomes:

$$\sum_{i=1}^{n}(R(x, y_i) \cdot \text{clip}(r(\theta_1, ref), 1 - \epsilon, 1 + \epsilon) - \beta) \log \pi_\theta(y_i|x), \tag{16}$$

and we first update $\theta$, then when we do the next iteration we set $\theta_1 = \theta$. We denote $\eta_i = R(x, y_i) \cdot \text{clip}(r(\theta_1, ref), 1 - \epsilon, 1 + \epsilon) - \beta$ and we can see that $\eta_i \approx R(x, y_i)$ when $\epsilon, \beta$ are small.

We write $\alpha_i = \frac{\eta_i}{\sum_{j=1}^{n} \eta_j}$, and since the denominator is independent of $x$, we just need to optimize

$$\mathcal{L}_{\text{R-LoL}} = \sum_{i=1}^{n} \alpha_i \log \pi_\theta(y_i|x), \tag{17}$$

and $\sum \alpha_i = 1$. By using the Lagrange's method, construct (for simplicity, write $z_i = \pi_\theta(y_i|x)$)

$$F(z_1, z_2, ..., z_n, \beta) = \sum_{i=1}^{n} \alpha_i \log z_i - \beta(\sum_{i=1}^{n} z_i - 1), \tag{18}$$

and the partial derivatives are

$$\frac{\partial F}{\partial z_i} = \frac{\alpha_i}{z_i} - \beta, \frac{\partial F}{\partial \beta} = -(\sum_{i=1}^{n} z_i - 1). \tag{19}$$

Hence we can see that the optimal solution of the A-LoL loss is: (BY solving the partial derivatives the optimal solution must have the same distribution as $\alpha_i$)

$$\pi^*(y_i|x)/\pi^*(\{y_1, ..., y_n\}|x) = \frac{R(x, y_i) \cdot \text{clip}(r(\theta_1, ref), 1 - \epsilon, 1 + \epsilon) - \beta}{\sum_{j=1}^{n}(R(x, y_j) \cdot \text{clip}(r(\theta_1, ref), 1 - \epsilon, 1 + \epsilon) - \beta)}. \tag{20}$$

So we can see that for close rewards responses, this optimal solution will not distinguish their distributions. For example, if we have two responses $y_1, y_2$ with $R_1, R_2$ as their rewards, then the A-LoL method will give $\frac{R_1}{R_1 + R_2}$ and $\frac{R_2}{R_1 + R_2}$ as the optimal solution and could be closed to $1/2$ if $R_1/R_2$ is closed to 1 (e.g. $R_1 = 100, R_2 = 99$). However, for our method, it will distinguished by $\exp(R_1 - R_2)$, i.e. for $R_1 = 100, R_2 = 99$, our method gives us $\frac{e}{e+1}$ and $\frac{1}{e+1}$, which seems better.

### A.2  Loss Bound Analysis

**Lower Bound of Positive Weighted Loss**

**Theorem 1.** *For any policy $\pi_\theta(\boldsymbol{y}|\boldsymbol{x})$ satisfying $\sum_y \pi_\theta(\boldsymbol{y}|\boldsymbol{x}) = 1$ and weights $w(\boldsymbol{x}, \boldsymbol{y}) > 0$, the weighted SFT loss satisfies:*

$$\mathcal{L}(\theta) = -\mathbb{E}_{x,y}\left[w(x, y) \log \pi_\theta(\boldsymbol{y}|\boldsymbol{x})\right] \geq 0, \tag{21}$$

*with equality if and only if $\pi_\theta(\boldsymbol{y}|\boldsymbol{x}) = \delta_{y=y^*}$ where $\delta_{y=y^*}$ is the optimal policy when $y^* = \arg\max_y w(x, y)$.*

*Proof.* Using the inequality $\log z \leq z - 1$ for $z > 0$:

$$\mathcal{L}(\theta) = -\mathbb{E}[w \log \pi_\theta] \tag{22}$$
$$\geq -\mathbb{E}[w(\pi_\theta - 1)] \quad (\text{since } \log \pi_\theta \leq \pi_\theta - 1) \tag{23}$$
$$= \mathbb{E}[w(1 - \pi_\theta)] \tag{24}$$
$$\geq 0 \quad (\text{since } w > 0 \text{ and } \pi_\theta \leq 1). \tag{25}$$

The equality holds when $\pi_\theta(\boldsymbol{y}|\boldsymbol{x}) = 1$ for the $y$ maximizing $w(x, y)$. $\qquad\square$

## B    Limitations and Future Work

Limited by training resources, the effectiveness of our method can only be validated on Qwen2.5-32B-Int4 at maximum, rather than on LLMs with higher precision or larger scales. Additionally, during experiments, we observed some inconsistent performance in LLMs after alignment with our method.

In the future, we will explore several promising directions to further enhance our framework. First, we aim to develop an online version of our method, enabling real-time interaction for calculating $Z(\boldsymbol{x})$ and updating the policy $\pi_\theta$ dynamically. Second, we plan to conduct extensive experiments across a broader range of tasks, such as multi-turn dialogue and long-form text generation, to validate the generalizability of our approach. Finally, scaling our framework to larger models and testing on more diverse and noisy preference datasets will provide deeper insights into its scalability and robustness.

## C    Case Study

Table 9 compares responses from SFT, SFT+VAR, SFT+DPO, Base+VAR, and Base+DPO, along with their corresponding reward scores. Notably, the SFT+DPO response is excessively verbose and includes irrelevant details, yet it receives the highest reward score. This highlights a tendency of the reward model to favor longer sequences, even when the content is less helpful or accurate. In contrast, SFT+VAR provides a concise and accurate answer, demonstrating the effectiveness of our method in balancing response quality and length. Additionally, Base+DPO generates a highly structured but overly complex and repetitive response, while Base+VAR delivers a clear and straightforward answer. Importantly, Base+VAR achieves performance comparable to SFT+VAR, showing that our method can effectively align models even when starting from the base (pre-trained only) version, without the need for SFT.

Table 9: Sample generations from different aligned versions of Llama3.1-8B for a prompt about checking a car's gas tank capacity.

| Prompt | How can I look up how much gas my car can hold? | Reward |
|---|---|---|
| **SFT** | You can find out how much fuel your vehicle holds by checking the owner's manual or looking on the door jamb (where some vehicles display this information). You may also be able to use an online resource, such as Edmunds.com, which has detailed specifications for most cars. | 31.36 |
| **SFT+VAR** | You can find out how much gas your car can hold by looking at the vehicle's owner's manual or contacting the manufacturer. The amount of gas your car can hold will vary depending on the make and model, as well as any modifications that have been made to it. | 31.69 |
| **SFT+DPO** | The total amount of gas that a car can hold will depend on the type and size of the car, as well as local environmental conditions and safety regulations. To find out exactly how much gas a specific car can hold, it is best to check with the car manufacturer or local fuel suppliers.
In general, larger cars such as 4x4s and vans can store more gas than smaller cars like compact and economy cars. Most modern vehicles have a fuel tank capacity of between 50-120 litres (depending on vehicle size), but some heavy-duty vehicles may have larger tanks of 120-130L or more.
It's also important to note that many countries have limits on how much LPG or CNG a vehicle can store due to safety concerns - for example, most countries limit the storage of LPG to no more than 20kg, while CNG is usually limited to 70m3 (for light duty vehicles) and 100 m3 (for heavy duty vehicles). If you're not sure what these limits are in your country, please check with the relevant authorities or consult the car manufacturer. In addition, certain areas may restrict the use of alternative fuels such as LPG and CNG during air pollution season or other times when air quality is deemed unhealthy ... | 60.75 |
| **Base+VAR** | You can lookup the capacity of your car's fuel tank by consulting your vehicle's manual or by contacting the manufacturer. Alternatively, you can estimate the capacity based on the size and shape of the fuel tank. | 28.58 |
| **Base+DPO** | Sure, there is an easy way to look this up:
1. Look up the cars make and model - This will give us its total capacity
2. Find its fuel tank size - This will tell us what type of vehicle it is
3. Check if it has any additional tanks or containers that may be used for gas
4. Calculate the vehicles maximum capacity
5. Add up any additional tanks or containers that may be used for gas
6. Subtract from the maxiumum capacity anything not yet accounted for
7. Finally, we'll add back any additional tanks or containers that were previously subtracted from
8. Finally, we'll add back any additional tanks or containers that were previously subtracted from
9. Finally, we'll calculate the final maximum capacity
10. We'll now want to check that our calculated maximum capacity makes sense.
Let's go through each step:
1: Make and model - To find out the vehicle's make and model (and year), let's run a Google search.
The results should include the following information about the vehicle:
- Make/model/trim level
- Fuel economy
- Transmission
- Cargo volume
- Total width
- Wheelbase
- Ground clearance ... | 52.33 |

