# OpenReview forum: "RLHF in an SFT Way: From Optimal Solution to Reward-Weighted Alignment"
_TMLR — Accepted by TMLR_

### Review · Reviewer_qm74 · 2025-09-14

**Summary Of Contributions:**

The paper proposes to use a variational approach to solve reinforcement learning with human feedback (RLHF) problem by minimizing the distribution difference between an optimized policy and the optimal solution to RLHF. The method named Variational Alignment with Re-weighting (VAR) provides a solution  takes the form of a reweighed supervised fine-tuning (SFT) . The paper claims that this solution is superior to DPO (an off-line baseline) and GRPO (online baseline) and provides empirical evidence to support the claim. The paper proposes to use a Monte Carlo-based estimator for the partition function to implement  VAR. Another interesting claim based on the analysis is that the reweighing values should be non-negative to ensure stability during optimization. Experimental evidence is provided to show the merits of this method

(EDIT: updated claims made to Yes after rebuttal)

**Audience:**

Yes

**Audience Explanation:**

RLHF-related methods is an active field so many folks in TMLR community will be interested in the work

**Broader Impact Concerns:**

Since this work uses RLHF, a statement similar to what other papers say on the impact of their method would be nice

**Claims And Evidence:**

Yes

**Claims Explanation:**

- The paper claims to be an offline alignment method like DPO and uses DPO as the baseline for comparison. However, Algorithm 1 listed in the paper calls for a reward model. Is this reward model explicit? If it's explicit then is the comparison to DPO fair?

- The method, VAR, does not outperform GRPO which is online and has an explicit reward function. If so, one conclusion is that the current method has limitations and online methods like GRPO/PPO should be preferred. This point needs to be addressed as a central claim is that regularized methods like clipping prevent the optimization process to reach RLHF minimum.

- The paper provides a Monte Carlo-like estimate for the partition function. The paper briefly studies the nature of the estimate in Section 4.5. Would it be possible to get more details on the variance of the estimates with batch sizes. Related to this point, the paper defines a term "micro-batch". What does this term mean?

- Another claim made in the paper is that the space of reward values should be non-negative. An exponential function is used as this follows from substituting the optimal value from DPO paper [Rafailov et al.]. However, values can be made non-negative by using other functions such as $f(x) = x^2$. Would it make sense to study this ablation to test the claim?

(I have some questions about the claim that may stem from my confusion due to being less familiar with some of the approaches in RLHF.  My  hope is that my questions will be clarified during the authors-reviewers process. Looking forward to authors response.)

**Requested Changes:**

- In Table 1 and 2, the paper uses several metrics. It would be helpful to the reader to know whether "higher is better" or "lower is better" to improve readability and interpretability

- Due to batching (micro-batching), does the run-time of the method increase over other methods?

- I was wondering about the claim on non-negative weights which happens naturally with the use of exponential(). Are there other functions that can work just as well? The reader is trying to figure out whether this argument is critical or just a consequence of using closed form solution to DPO

---

> ### Author Response · Authors · 2025-10-27
> **Revised Manuscript Link**
>
> We sincerely thank the reviewers for their valuable feedback. We have uploaded our revised manuscript, which addresses their comments, to the following anonymous link for your review: https://anonymous.4open.science/r/TMLR_VAR_revision1-8FA3/TMLR_VAR_revision.pdf

---

> ### Author Response · Authors · 2025-10-27
> **Response to Weaknesses and Requested Changes**
>
> 1. **Response to Weakness 1: Reward Model and DPO Fairness**:
>
> We appreciate the reviewer’s question on comparing VAR (which uses an explicit reward model) with DPO.
>
> VAR’s reward model is indeed explicit, but it is trained directly from human preference data—just like the implicit preference signals DPO relies on. Specifically, the reward model we used (e.g., OffsetBiasRM in Section 4.1) is learned using human preference datasets (e.g., OffsetBias, which contains "chosen" and "rejected" response pairs similar to DPO’s required data), meaning the core human preference source for both VAR and DPO is consistent.
>
> Therefore, we consider this a fair comparison.
>
>
> 2. **Response to Weakness 2: VAR (Offline) vs. GRPO (Online) Performance**:
>
> We thank the reviewer for raising the comparison between VAR (offline) and GRPO (online).
>
> On performance, VAR achieves results comparable to GRPO on Qwen2.5‑3B (Avg. All: 67.25 vs. 68.23 in Table 4) and **outperforms** GRPO on Llama3.2‑3B (61.51 vs. 60.34). Notably, GRPO needed SFT initialization in that experiment for stable training, whereas VAR worked directly from the base model. In addition, VAR trains **over 5$\times$ faster** per epoch (e.g., 42 min vs. 2 h 54 min for Qwen2.5‑3B) and avoids the heavy cost of online sampling.
>
> We acknowledge the limitation that, in our experiments, the effective sample size per batch was relatively small due to resource constraints. Our theoretical derivation shows that with a sufficiently large batch size $B$, the approximation to the RLHF optimum $\pi^*$ would be even tighter, and performance can improve further. We have not yet been able to validate this upper bound experimentally due to computational limitations, but the theoretical guarantee remains.
>
> Regarding clipping in GRPO/PPO: our point is not that these methods fail to align effectively, but that their clipping adds bias—it flattens fine reward differences (e.g., treating scores 100 and 99 almost the same). GRPO uses this as a trade‑off for stability in an online setting; by contrast, VAR is designed for efficiency in an offline setting. We use non‑negative, reward‑based weights from the closed‑form RLHF solution and avoid clipping altogether. This removes the bias, keeps the optimization landscape well‑bounded, and directly targets $\pi^*$.
>
> In scenarios with abundant compute, online methods like GRPO/PPO remain strong candidates. In most practical settings, however, VAR offers a different—and deliberate—trade‑off: comparable or better performance at a fraction of the cost, stable optimization without clipping, and potential for even higher performance when batch size can be scaled.
>
> 3. **Response to Weakness 3: $\hat{Z}(x)$ Estimator Variance and "Micro-Batch" Clarification**:
>
> We thank the reviewer for the questions about our partition function estimation and the definition of "micro-batch". These clarifications will indeed help improve the paper.
>
> **On "micro-batch"**: You're right to ask for a clear definition. In our work, a **micro-batch** simply refers to the small set of training samples processed at once (e.g., $B=8$). This term is common in LLM training, where a large "global batch" is often split into smaller micro-batches that fit on a GPU.
>
> In our case, this micro-batch serves **two purposes**:
>
> 1.  It's used for the standard [SFT-like] gradient update.
> 2.  It provides the set of samples for our Monte Carlo estimation of the partition function $\hat{Z}(x)$ (as shown in Equation 9 and Algorithm 1).
>
> **On the variance of ($\hat{Z}(x)$) estimates:** We agree that more detail on the variance is helpful. We've added a new analysis in **Section 4.6 (Figure 4)** to address this directly. This new figure plots the variance of our $\hat{Z}(x)$ estimate against the micro-batch size $B$ for two different methods:
>
> 1.  **Offline (In-Batch) Estimation (Fig. 4a):** This is the efficient method **we use in our paper**. It uses the samples *already in the micro-batch* for the estimation. As the plot shows, the variance **drops significantly** as $B$ increases from 2 to 8, confirming our estimate gets much more stable with a modest batch size.
> 2.  **Online Sample Estimation (Fig. 4b):** We added this **for comparison**. This method would require sampling $B$ *new* responses from $\pi_{ref}$ just for the estimation.
>
> This comparison clearly shows that our in-batch (offline) approach is not only more computationally efficient but also demonstrates a stable and consistent reduction in variance as the batch size grows. We believe this new **Section 4.6** and **Figure 4**  better support our methodological choice.

---

> > ### Author Response · Authors · 2025-10-27
> > **Response to Weaknesses and Requested Changes**
> >
> > 4. **Response to Weakness 4 and Required Changes 3: Justification of the Exponential Transformation**：
> >
> > We sincerely thank the reviewer for this insightful question. It touches upon the theoretical foundation of our method and highlights a key aspect of our work.
> >
> > The reviewer is correct that $f(x)=x^2$ would also ensure non-negative weights. However, our use of the exponential function is **not an arbitrary design choice** for stability; it is a **mathematical requirement** derived directly from the standard RLHF objective.
> >
> > Our method, VAR, works by minimizing the KL divergence to the known closed-form optimal policy, $\pi^{\*}$. This optimal policy is derived from the standard **Bradley-Terry (BT) preference model**, and as shown in prior work [Rafailov et al.] and our **Equation 3**, it has a specific exponential form: $\pi^{\*}(y|x) \propto \pi_{ref}(y|x)\exp(\frac{1}{\beta}r(x,y))$. When we derive our VAR loss, this $\exp(\cdot)$ weighting function is the **direct and only theoretical result** of this derivation.
> >
> > Therefore, using a different function like $f(x)=x^2$, while positive, would unfortunately **break this theoretical link**, where we would no longer be optimizing towards the *true* optimal solution defined by the BT model.
> >
> > A crucial benefit of this theoretically-grounded $\exp(\cdot)$ function is that it *also* solves the instability problem. As we discuss in **Section 3.3**, negative weights ($-\log \pi_\theta$ toward $+\infty$) make the training process unstable. Our derived $\exp(\cdot)$ weight is guaranteed to be strictly positive, ensuring a stable, bounded optimization by construction.
> >
> > That said, the reviewer raises a very important point. Our derivation is fundamentally tied to the Bradley-Terry model. It is plausible that if one were to adopt a *different preference model* (e.g., as explored in [Wang et al.]), the resulting closed-form optimum might indeed take a different, non-exponential form. Exploring such alternatives is a valuable direction for future work. For this paper, however, our goal was to provide a stable and efficient solution for the standard BT-based RLHF problem, which necessitates the exponential form.
> >
> > References:
> >
> > [Rafailov et al.] Direct Preference Optimization: Your Language Model is Secretly a Reward Model
> >
> > [Wang et al.] Beyond reverse KL: Generalizing direct preference optimization with diverse divergence constraints.
> >
> >
> > 5. **Response to Requested Changes 1**:
> >
> > We apologize for not clarifying the metric direction. We have added a clear note to all tables in the revision. Thank you for this suggestion to improve clarity.
> >
> > 6. **Response to Required Changes 2**:
> >
> > Thank you for raising this important question about the practical runtime efficiency.
> >
> > Our method is structured to avoid adding significant overhead to the main training loop. We achieve this by integrating the costliest parts of the $\hat{Z}(x)$ estimation into the **data preprocessing step**:
> >
> > 1.  **Preparation:** Before the main training loop starts, we pre-shuffle the training data and split it into the exact **micro-batches** ($B=8$) that will be used for training.
> > 2.  **Reward Calculation:** We then calculate the reward $r(x,y)$ for each in-batch pair and store these values on disk.
> >
> > Therefore, during training, we only need to perform one major extra step compared to standard SFT: a forward pass through the **reference model ($\pi_{\mathrm{ref}}$)** to obtain the logits necessary to compute $\hat{Z}(x)$ (using the pre-calculated rewards).
> >
> > However, the most time-consuming part of training LLMs is the **backward pass (gradient computation)**. Since our training loop does not involve running the backward pass on the reference model ($\pi_{\mathrm{ref}}$ is frozen), the minimal extra cost incurred by the single forward pass is negligible when amortized over the entire training process.
> >
> > We have added a comparison of SFT, DPO, and VAR under the same settings (GPU, batch size, same training samples) in Table 8 of the revised manuscript. The results show that the training time for VAR is close to SFT and DPO, confirming that our approach is highly efficient.
> >
> > 7. **Response to Broader Impact Concerns**:
> >
> > We thank the reviewer for this helpful suggestion. In the revised manuscript, we have added an **Impact Statement** section that highlights the potential impact of our method within the broader context of RLHF research.

---

> > > ### Comment · Reviewer_qm74 · 2025-10-31
> > > **Thank you**
> > >
> > > I thank the authors for the detailed rebuttal. The information provided is sufficient for me to get a better understanding of the paper. I am satisfied with the rebuttal and can now see that updated paper has support for the claims made in the paper. I am in favor of accepting this paper provided the authors commit to making all updates included in their rebuttal response.
> > > I have no further questions.
> > >
> > > Finally, I would like to bring the following concurrent work to the authors' attention. I would like to understand if the paper might be included as related work in the draft.
> > >
> > > "On the Generalization of SFT: A Reinforcement Learning Perspective with Reward Rectification" https://arxiv.org/abs/2508.05629

---

> > > > ### Author Response · Authors · 2025-10-31
> > > >
> > > > Thank the Reviewer qm74 for your comprehensive and constructive suggestion to help complete our work. We will polish our writing to help make for smoother reading, and will add your provided paper into the discussion in our final version.

---

### Review · Reviewer_Di1a · 2025-09-28

**Summary Of Contributions:**

This paper introduces a novel framework that recasts Reinforcement Learning from Human Feedback (RLHF) as a variational inference problem. The core objective is to directly minimize the KL divergence between the learning policy and the known closed-form optimal solution of the RLHF objective. This reformulation yields a new alignment algorithm, Variational Alignment with Re-weighting (VAR), which manifests as a simple, reward-driven weighted Supervised Fine-Tuning (SFT) loss. The weights are derived via an exponential transformation of the reward, guaranteeing them to be non-negative. This design resolves the training instabilities that can arise in methods like DPO.

**Audience:**

Yes

**Audience Explanation:**

This is a good read. The motivation is solid, the proposed solution is elegant, and the writing is exceptionally clear. The VAR method, which reframes the complex RLHF problem into a stable, weighted SFT objective, is a commendable contribution. I must miss something or this paper should be accepted, though I have one primary concern that I would like the authors to address.

My main reservation regards the given SFT-based formulation. Recent literature (Sft memorizes, rl generalizes: A comparative study of foundation model post-training; Reinforcement Learning Finetunes Small Subnetworks in Large Language Models) has highlighted the critical role of reinforcement learning paradigms for better generalization within alignment. Since the proposed VAR objective is ultimately a weighted SFT loss, it is crucial to assess whether it inherits the potential generalization limitations of SFT, even if it outperforms in-domain.

To this end, I would like to see a more convincing evaluation of the model's out-of-domain, conversational abilities. The paper would be significantly strengthened by the inclusion of results on widely-recognized, general-purpose benchmarks such as AlpacaEval 2.0 and Arena-Hard 0.1. To be clear, my expectation is not that VAR must outperform DPO on these benchmarks; rather, demonstrating that it achieves competitive or reasonably close performance would be sufficient to alleviate this concern.

**Broader Impact Concerns:**

No.

**Claims And Evidence:**

No

**Claims Explanation:**

The authors have conducted detailed comparisons of the proposed VAR method against baselines like DPO across two model families (Llama and Qwen (why we are still using Qwen2.5? which is commonly known as a questiable model)) of various sizes. The reward score and win-rate evaluations on the HHA benchmark, in particular, provide strong support for the paper's core claims regarding VAR's stability and effectiveness as an optimization algorithm.

Nevertheless, there are two areas in the evaluation section where significant improvements could be made:

The current evaluation focuses predominantly on the HHA benchmark. While this setup is well-suited for demonstrating the algorithm's effectiveness in optimizing a given reward function (an in-domain evaluation), it does not capture the general conversational and instruction-following capabilities. To substantially strengthen the claims about VAR's practical utility, I strongly recommend including results on at least one widely-recognized, automated conversational benchmark, such as AlpacaEval 2.0 or Arena-Hard 0.1.

In the win-rate evaluation, the paper states it "adopt the evaluation template from MT-Bench" while reporting results "on the HHA benchmark." This phrasing could be confusing to readers, potentially leading them to believe the standard MT-Bench benchmark was run. I suggest the authors explicitly clarify in the experimental setup that they are applying the judging protocol from MT-Bench (i.e., the GPT-4 judge with its standardized prompt template) to evaluate model outputs generated from prompts from the HHA test set. (Why not judged by MT-Bench benchmark?)

**Requested Changes:**

The paper's motivation for avoiding instabilities present in methods like DPO is well-founded. However, the description of DPO's failure mode as ``employing negative weights'' is an over-simplification. The cited works (3D-Properties: Identifying Challenges in DPO and Charting a Path Forward; Smaug: Fixing Failure Modes of Preference Optimisation with DPO-Positive) offer a more nuanced explanation based on imbalanced or incorrect gradient dynamics, rather than an explicit negative weighting scheme.

For example, stating like (just for toy) ``the gradient descent on dis-preferred responses can become pathologically strong, leading to an unstable optimization landscape'' would be more accurate.

---

> ### Author Response · Authors · 2025-10-27
> **Revised Manuscript Link**
>
> We sincerely thank the reviewers for their valuable feedback. We have uploaded our revised manuscript, which addresses their comments, to the following anonymous link for your review: https://anonymous.4open.science/r/TMLR_VAR_revision1-8FA3/TMLR_VAR_revision.pdf

---

> ### Author Response · Authors · 2025-10-27
> **Response to Weaknesses and Requested Changes**
>
> 1. **Response to Weakness 1: Generalization Benchmarks and Qwen Model Choice**:
>
> We thank the reviewer for the positive feedback and helpful suggestions.
>
> On the use of Qwen2.5: We selected Qwen2.5 because it is a stable non-thinking version that allows controlled comparison. In contrast, Qwen3 mixes thinking and non-thinking modes, which introduces uncontrollable variations during inference and makes a fair evaluation of optimization methods difficult under our limited resources.
>
> On broader evaluation: Following the reviewer’s recommendation, we added results on AlpacaEval 2.0 and Arena-Hard 0.1 (see Table 6 in the revised manuscript), where VAR performs on comparable or better than DPO on both benchmarks, confirming that its stability and effectiveness generalize beyond the HHA setting.
>
> 2. **Response to Weakness 2: Clarification on Evaluation (MT-Bench and HHA)**:
>
> We thank the reviewer for this helpful clarification. We have updated the experimental setup to explicitly state that our win-rate evaluation uses the GPT-4 judge with its standardized prompt template from MT-Bench, applied to responses from the HHA test set, rather than running the MT-Bench benchmark itself (See page 6 Evaluation part of the revised manuscript).
>
> We appreciate the reviewer’s question on why MT-Bench itself was not used. Our HHA experiments are designed to evaluate models trained and tested within the HHA dataset; therefore, we kept the prompts consistent with this setup. We adopted the MT-Bench judging protocol only to ensure comparability under a widely recognized evaluation standard.
>
> For general conversational and instruction-following benchmarks similar to MT-Bench, we have also included results on four public benchmarks (Table 6) and, following the reviewer’s earlier suggestion, added AlpacaEval 2.0 and Arena-Hard 0.1. These results confirm that VAR’s stability and effectiveness generalize beyond the HHA setting.
>
> 3. **Response to Requested Changes 1**:
>
> We thank the reviewer for the helpful clarification and suggested wording. We have revised the description accordingly to reflect that the instability arises from imbalanced or incorrect gradient dynamics (see page 5 around equation 12).

---

> > ### Comment · Reviewer_Di1a · 2025-11-07
> > **We thank the authors for their rebuttal.**
> >
> > The new experiments on AlpacaEval 2.0 and Arena-Hard 0.1 have addressed my primary concern. The clarification on the HHA/MT-Bench evaluation protocol and the more precise description of DPO's instability mechanism are also satisfactory.
> >
> > As a further suggestion, the authors identify a limitation of methods like DPO as the potential for instability as a motivation.
> > This specific issue, stemming from DPO's reliance on a fixed reference model ($\pi_{\text{ref}}$), is a highly active area of current research. Several recent works have proposed different solutions to this exact problem.
> >
> > I would suggest considering a brief discussion of these emerging strategies in related work, for instance:
> >
> > - Some approaches now avoid the reference model entirely (Simpo: Simple preference optimization with a reference-free reward).
> >
> > - Some approaches focus on learning a more suitable reference model (Learn your reference model for real good alignment).
> >
> > - And others aim to correct the distributional mismatch between the reference and the policy (Mitigating Mismatch within Reference-based Preference Optimization).
> >
> > Briefly situating VAR in the context of these recent developments would underscore its unique contribution to solving DPO's known limitations.
> >
> > I will be submitting my final recommendation for acceptance.

---

> > > ### Author Response · Authors · 2025-11-08
> > >
> > > Thank the Reviewer Di1a for your comprehensive and constructive suggestion to help complete our work. We will polish our writing to help make for a smoother reading and more accurate description, and will add your provided paper into the discussion in our final version.

---

### Review · Reviewer_9EjE · 2025-10-02

**Summary Of Contributions:**

**Strength**

Proposes Variational Alignment with Re weighting (VAR), an offline, reward weighted SFT objective derived by minimizing the forward KL between the RLHF closed form optimum ( $\pi^* $ ) and the learned policy ($ \pi_\theta$ ). The paper derives ( $\pi^* $ ) (Eq. 3) and shows that maximizing ($ \mathbb{E}{\pi^*}[\log \pi\theta] $) yields a reward weighted SFT loss (Eq. 8–10), implemented via Algorithm 1 with an in batch estimator for the partition function (Z(x)).

Argues theoretically that VAR avoids (i) clipping bias in PPO/ALoL and (ii) instability from negative weights (e.g., DPO/ALoL), by operating over positive measures ($w(x,y)\propto \pi_\mathrm{ref}(y|x)\exp(r/\beta)>0$) (Sec. 3.3; App. A.1–A.2).

**Weakness**

Partition Estimation is not accurate. The current in batch estimator (Sec. 3.2; Algorithm 1) reuses candidate responses across prompts; without a bias analysis or ground truth comparison, it is unclear how accurate ( $\hat Z(x)$ ) is, and cross prompt reuse may systematically mis estimate normalization for some prompts.

Time and memory comsumption is not addressed. Beyond Table 4’s GRPO comparison, the paper lacks a direct VAR vs. DPO compute/memory analysis. Given the ($O(B^2)$) cross prompt scoring in Algorithm 1, clearer profiling (tokens/s, peak memory, throughput vs. sequence length) is needed to substantiate efficiency claims and guide practical adoption.

**Audience:**

Yes

**Audience Explanation:**

RLHF simplifications and offline preference optimization (e.g., DPO/ALoL/KTO) is a heated topic. A method that (i) ties directly to the RLHF optimum via a variational view, (ii) works as a drop in, reward weighted SFT loss, and (iii) demonstrates competitive performance and stability across widely used model families is highly relevant to both researchers and practitioners concerned with scalability and engineering simplicity of alignment. The paper’s clear comparisons on HHA and standard reasoning/code benchmarks (Tables 1–5; Fig. 1–2) make the findings immediately actionable.

**Claims And Evidence:**

Yes

**Claims Explanation:**

Effectiveness vs. DPO: On HHA, VAR outperforms DPO in average reward on nearly all model sizes and initializations (Tables 1–2), and achieves higher GPT 4o win rates on most Llama and Qwen models (Fig. 1; Table 3). These address the paper’s central claim that VAR is a stronger offline alternative.

Stability: Validation reward curves show DPO’s volatility versus VAR’s steady increase and convergence (Fig. 2), directly supporting the “training stability” claim.

Efficiency vs. online RL: The head to head with GRPO includes wall clock times per epoch and shows VAR achieves comparable rewards while being ~4–5× faster per epoch, depending on the model (Table 4). This supports the “significant computational savings” message in the abstract with concrete numbers.

**Requested Changes:**

1.Quantify the accuracy of the partition estimator

Provide a bias/variance analysis of the in batch Monte Carlo estimator in Sec. 3.2, especially because Algorithm 1 forms ($ \hat Z(x_i) $) using responses ($y_j$) drawn under other prompts ($x_j$) but re weighted under ( $\pi_\mathrm{ref}(y_j|x_i)$ ). This cross prompt reuse is convenient but may be distribution mismatched relative to the ideal (y\sim\pi_\mathrm{ref}(\cdot|x_i)). Include a sanity study: for a subset of prompts, estimate ($Z(x)$) with many samples ($y\sim\pi_\mathrm{ref}(\cdot|x)$) and report the relative error of ( $\hat Z(x) $) as a function of (B). (Sec. 3.2; Eq. 9–10; Algorithm 1.)

2.Report full compute profiles and complexity vs. DPO
While Table 4 compares against GRPO, please also report wall clock time/epoch, tokens/s, and peak GPU memory for VAR vs. DPO under identical hardware, batch size, and sequence lengths. Given Algorithm 1’s ($O(B^2)$) cost for cross prompt reference logits and reward evaluations (steps 3–5), quantifying actual overheads will clarify when VAR is faster/slower than DPO in practice. (Algorithm 1.)

3.Ablations that isolate design choices of ( $\hat Z(x)$ )
Compare the current cross prompt estimator to (a) within prompt candidates sampled from ( $\pi_\mathrm{ref}(\cdot|x_i)$), (b) top k candidate subsets, or (c) cached candidate pools built per prompt across epochs. Report effects on accuracy, speed, and memory. (Sec. 3.2; Algorithm 1.)

---

> ### Author Response · Authors · 2025-10-27
> **Revised Manuscript Link**
>
> We sincerely thank the reviewers for their valuable feedback. We have uploaded our revised manuscript, which addresses their comments, to the following anonymous link for your review: https://anonymous.4open.science/r/TMLR_VAR_revision1-8FA3/TMLR_VAR_revision.pdf

---

> ### Author Response · Authors · 2025-10-27
> **Response to Weaknesses and Requested Changes**
>
> 1. **Response to Requested Changes 1**:
>
> We thank the reviewer for this insightful comment regarding the accuracy of the partition estimation. We agree that a more detailed analysis is necessary to understand the bias and reliability of the in-batch estimator ($\hat{Z}$(x)).
>
> In the revised manuscript, **Figure 4** has been added to further explore this issue. Specifically, we analyze the variance of ($\hat{Z}$(x)) under different micro-batch sizes (B), comparing:
>
> 1. **Offline estimation** – our in-batch approach as described in Section 3.2 and Algorithm 1.
> 2. **Online sampling estimation** – where samples are drawn independently from ($\pi_{\mathrm{ref}}$) for each prompt.
>
> The results (both for 50 samples) show that while cross-prompt reuse introduces slight bias for small (B), the estimator quickly stabilizes as (B) increases, confirming that the in-batch approximation provides a reliable and efficient normalization in practice. We appreciate the reviewer’s suggestion, which helped us strengthen this part of the analysis.
>
> 2. **Response to Requested Changes 2**:
>
> Thank you for this excellent suggestion. We agree that quantifying the computational overhead is very important.
>
> We have added a new **subsection "Computational Overhead Analysis" (Section 4.7)**, including **Table 8**, to directly address this. This new section provides a detailed comparison of wall clock time, throughput (tokens/s), and peak GPU memory for VAR versus SFT and DPO under identical settings.
>
> The results confirm that VAR's practical overhead is minimal. As explained in the new section, we pre-calculate rewards, and the main training bottleneck is the backward pass (gradient computation). Since VAR only adds a forward pass for the frozen $\pi_{\mathrm{ref}}$ model, the extra cost is negligible. Our empirical results show that VAR's training time is close to DPO and SFT, confirming its high efficiency.
>
> We believe this new analysis and table fully clarify the practical efficiency of our method.
>
> 3. **Response to Requested Changes 3**:
>
> Thank you for this excellent suggestion to further ablate the $\hat{Z}(x)$ estimator.
>
> To address your core question (a), we have added a new baseline to **Table 7 (Section 4.5)**, labeled **$8^{\*}$**. This method estimates $\hat{Z}(x)$ using dedicated candidates pre-sampled from $\pi_\text{ref}$, contrasting with our efficient in-batch method.
>
> The results in the updated Table 7 show this $\pi_\text{ref}$-sampled approach is not only **more computationally expensive** (requiring a full, separate pre-sampling pass) but also resulted in **slightly lower performance** (68.32 Avg. vs. **68.42** Avg.). This finding validates our design choice of using the simpler, more efficient in-batch estimator. We have updated **Section 4.5** with this analysis.

---

### Public Comment · ~Chunjin_Jiang1 · 2025-09-26
**About the evaluation indicators of HHA - reward score**

I greatly admire your excellent article and the breadth of your work. I have a few questions that I could not find answers to in the paper. Specifically, regarding the metrics reported in Table 1, could you clarify how they were computed? Since the reward scores output by NCSOFT/Llama-3-OffsetBias-RM-8B are unbounded real numbers, how did you map them to a 0–100 scale? Would it be possible to provide more explanation, or to consider using more widely adopted evaluation metrics?

---

> ### Author Response · Authors · 2025-10-27
> **Response to public comment**
>
> Hi, thanks for your interest in our work.
> You are correct, the raw reward scores are unbounded. To map them to the 0-100 scale reported in the tables, we simply apply a Sigmoid function to the raw score and then multiply by 100.
>
> **Scaled Reward = Sigmoid(Raw Reward Score) $\times$ 100**

---

### Decision · Action_Editor_MVB8 · 2025-12-19

**Recommendation:** Accept with minor revision

**Additional Comments:**

- Finalize all rebuttal-added content in the paper: keep the added AlpacaEval 2.0 / Arena-Hard 0.1 results and describe the evaluation setup clearly.
- Partition estimator validation: include the added analysis/figure on variance/bias vs. micro-batch size and connect it directly to the Algorithm.
- Retain the new section/table comparing VAR vs. DPO/SFT with wall-clock time, tokens/s, peak GPU memory under matched settings; clearly state what is precomputed offline (e.g., rewards).
- Explicitly state you use the MT-Bench judge template/protocol on HHA prompts, and did not run MT-Bench itself

**Audience:**

Yes

**Audience Explanation:**

Offline alignment / preference optimization and RLHF simplification remain highly active topics, and this paper offers a clean variational perspective that yields a practical reward-weighted SFT-style objective with an emphasis on stability. Reviewers indicate the method and empirical comparisons are relevant and actionable for both researchers and practitioners working on post-training, and the revised manuscript broadens evaluation beyond the in-domain HHA setting to more general conversational benchmarks, increasing its interest to the broader TMLR readership.

**Claims And Evidence:**

Yes

**Claims Explanation:**

Across reviews and the rebuttal/revision cycle, the core claims appear well supported.

Reviewers highlight convincing empirical evidence that VAR is a strong offline alignment alternative, particularly due to the improvements vs. DPO on HHA reward/win-rate metrics, the stability via smoother training curves, and a clear wall-clock comparison vs. GRPO showing substantial per-epoch speedups.

There are a few main technical concerns raised by reviewers, including accuracy/variance of the partition estimator, compute/memory overhead vs. DPO, and ambiguity in the HHA + MT-Bench-judge evaluation wording. Most of them were explicitly addressed in the revision with additional analyses/tables/clarifications. A prior key concern about out-of-domain conversational generalization was also addressed with added AlpacaEval 2.0 and Arena-Hard 0.1 results, which the concerned reviewer confirmed alleviated their reservation and led them to recommend acceptance. Overall, the remaining items are primarily presentation/positioning polish, not issues that undermine soundness.